



# Quantifying Tropical Cyclone-Generated Waves in Extreme-Value-Derived Design for Offshore Wind

McElman Sarah[1,2], Verma Amrit Shankar[1], and Goupee Andrew[1]

[1]University of Maine, Orono, ME, USA
[2]Avangrid Renewables, Boston, MA, USA

**Correspondence:** McElman Sarah (sarah.mcelman@maine.edu)

**Abstract.** Wave extreme values, such as significant wave height, peak period, and crest height, are central to design and operation practices for offshore wind structures. However, the most suitable methods for deriving these extremes, both statistically and from numerical models, is not straightforward. This is especially acute in mixed-type climates, as in the Atlantic coast of the US, where tropical cyclones (hurricanes) and extra-tropical cyclones (winter storms) occur at the same locations with varying frequency and intensity. Limited guidance is provided in major offshore wind energy standards for the minimum requirements of these ocean models and methods used for determining accurate design and operational metocean conditions for regions with tropical cyclones and mixed-type environments. This study investigates the representation of extreme significant wave heights on the US Atlantic coast generated by mixed storm types, as represented in numerical simulations and univariate extreme value analysis. Notable differences between N-year design values are found, as projected by the two different modeled conditions with both block maxima and peaks-over-threshold methods. Attributing factors include hindcast duration, proximity of design location to historical track storm centers, and single analysis of mixed-type distributions. This paper is the first of its kind to propose a methodology for defining extreme significant wave heights due to tropical cyclones for offshore wind design and operation in Mid- and North-Atlantic waters. Recommendations for achieving accurate and representative extreme values for offshore design on the US Atlantic coast are provided.

## 1 Introduction

As offshore wind development expands to locations with mixed storm types, such as a combination of tropical and extra-tropical cyclones, new meteorological and oceanographic features influence infrastructure planning, design, and operation. The successful design and operation of offshore projects require long-term metocean data, traditionally in the form of a locally-validated, high-fidelity multi-decade hindcast of coupled winds, waves, currents, and water levels. These models are typically forced with or derive boundary conditions from global or downscaled reanalysis data sets, such as Climate Forecast System Reanalysis (CFSR) or European Climate Mid-Range Weather Forecast Reanalysis v5 (ERA-5) (e.g., as in Groll and Weisse (2017)). Data from these models form the basis for Extreme Value Analysis (EVA), the statistical determination of an N-year parameter. The results of these methods can be sensitive to method choice, parameterization, or data fit (Haselsteiner and Thoben (2020)).



The representation of coastal extreme events by numerical models are also sensitive to parameterization choices and model design decisions. Investigations in the literature have quantified tropical cyclone features and their influence on offshore wind design on turbine-scale dynamics. Kim and Manuel (2019) simulated the local wind, waves, and hydrodynamic features of Hurricane Sandy at a number of offshore wind development areas on the Atlantic coast with the Miami Coupled Model. They recommend coupled wind-ocean modeling for the best representation of the features and evolution of tropical cyclones as a

prerequisite to intensity assessment and wind-wave probability distribution. Gomez et al. (2023) quantified the convective momentum transfer of Category 1, 2, and 3 tropical cyclones, developed from a catalog of synthetic events on the US East Coast, and found that the resulting turbulence and gust characteristics of these scenarios at times exceeded current IEC standards.

Fewer studies have focused on tropical cyclone-generated waves and wave growth and how they are represented in offshore wind engineering decisions. Additionally, limited guidance is provided in major offshore standards (API RP 2MET, 2019; DNV,

2018; IEC–614000–3, 2019) for the minimum requirements of ocean models and methods for capturing tropical cyclone-generated N-year wave heights and periods. As a result, many different approaches have been taken to model and quantify these ocean design values. Few studies have investigated how current methods treat mixed climates with annual tropical and extra-tropical cyclone events. In the absence of abundant buoy observations in the path of a tropical cyclone at the points of interest, the regionally-validated GROW-Fine East Coast model is referenced as the ground truth to assess the results of current

methods–modeling and statistical analysis–in capturing ocean design parameters.

In this study, the selection of proper design values depends on accuracy in three tiers: statistical methodology, model capacity, and model design. Statistical characterization of the meteorological and oceanographic extremes on the US Atlantic Coast with General Extreme Value and Generalized Pareto methods are frequently selected in the literature for similar applications, for example in Haselsteiner and Thoben (2020), Northrop et al. (2017), Barthelmie et al. (2021), and Bhaskaran et al. (2023). In a

broader study, Kresning et al. (2024) investigated uncertainty in loads assessment for offshore wind on the US Atlantic coast resulting from a variety of univariate and bivariate extreme value methods, finding differences of up to 6 % between calculated return values. Neary et al. (2020) found that discrepancies between univariate and bivariate methods based on the same data were random and indicated "reasonable agreement on average".

The capacity of spectral wave models to resolve tropical storm features has been investigated by MacAfee and Wong (2007),

O'Grady et al. (2022), and Padilla-Hernández et al. (2007), and have shown that Simulating Waves Nearshore (SWAN) and the Wave Modeling Project (WAM), the basis for wave models in the GROW-Fine East Coast hindcast, are capable of capturing peak wave values and trapped-fetch swell generated by tropical cyclones. Additionally, a comparison of the performance of SWAN and MIKE-21 in coastal Portugal by Fonseca et al. (2016) has shown similar overall behavior of wave growth and propagation.

In addition to tool capacity, modeling choices such as boundary condition quality must be investigated for sufficient ability to resolve tropical cyclone features. As in Campos et al. (2022) and Gandoin and Garza (2024), metocean models frequently employ the ERA-5 reanalysis dataset, for direct analysis, or as a boundary condition to high-resolution modeling. Campos et al. (2022) and Gandoin and Garza (2024) found that ERA-5 under-captured peak winds during storm events, while Caires and Sterl (2005) and Stephens and Gorman (2006) found that ERA-based wave models under-captured significant wave heights





(Hs). Similarly, Neary et al. (2020) investigated 50-year significant wave height return values on the Atlantic coast and found a "systematic underbias for extreme significant wave height derived from model hindcasts as compared to those derived from buoy measurements." Each author suggests additional calibration for improved performance.

This paper investigates the performance of two high-resolution metocean models forced by global ocean reanalysis datasets to (1) represent significant wave height due to tropical and extra-tropical events, and (2) quantify differences in return period

values between these high-resolution models and datasets generated from direct modeling of tropical and extra-tropical storm events. Two locations on the US east coast are considered, which experience tropical and extra-tropical events with varying frequencies and magnitudes: in the North Atlantic ("NA", New England Wind) and in the Mid-Atlantic Bight ("MAB", Kitty Hawk Wind), to assess sensitivity of the analysis to different mixed-type climates (see map, Figure 1). A schematic of the study method is provided in Figure 2. Wind and ocean parameterization of all three models are documented in Table 1 and Table 2,

respectively. To ensure that the statistical findings from this study are not significantly sensitive to the selected extreme value method employed, a subset of model results are additionally analyzed with an alternative extreme value method and evaluated.

## 2    Methods

Two validated and calibrated high-resolution models of wind, waves, and hydrodynamics are used to simulate a multi-decade hindcast of hourly conditions around the two wind project areas (hereafter referred to as the North Atlantic and Mid-Atlantic

"high-resolution" models). Post-calibrated model results are compared to buoy observations and the reconstructed tropical and extra-tropical storm models (collectively referred to as the "GROW-Fine East Coast" model). Results are presented as a collection of absolute and normalized values to protect intellectual property of the GROW-Fine data, where necessary.

Return period results from all four models (high-resolution models in the North Atlantic and in the Mid-Atlantic Bight, the GROW-Fine tropical-cyclone-only model, and the GROW-Fine extra-tropical-cyclone-only model) are calculated by the Block

Maxima (BM) method with a Gumbel distribution fit.

To ensure that the results were not significantly influenced by the chosen statistical method, a sensitivity analysis of a subset of model results was carried out with Peaks-Over-Threshold (POT), using both 2-parameter Weibull and Gumbel distribution fits. A sample of results from this analysis is available in Appendix A. For both the North Atlantic (Figure A1a) and Mid-Atlantic Bight (Figure A1b) cases, fits by Weibull or Gumbel resulted in 4 % or less variation between return values in all

model subsets, except for the tropical 10,000-year MAB significant wave height (+8 %, Weibull). Larger variation was observed between return values by POT and BM analyses; however, the trends between models and storm types were consistent. Following Barthelmie et al. (2021) and Bhaskaran et al. (2023), the block maxima method was considered suitable for this study. To enable long hindcast periods for the Tropical and Extra-Tropical GROW-Fine models, only storm events are represented in these model datasets. As the dataset does not include normal sea states, it is not possible to determine extreme values

for this model by the Peaks-Over-Threshold method.

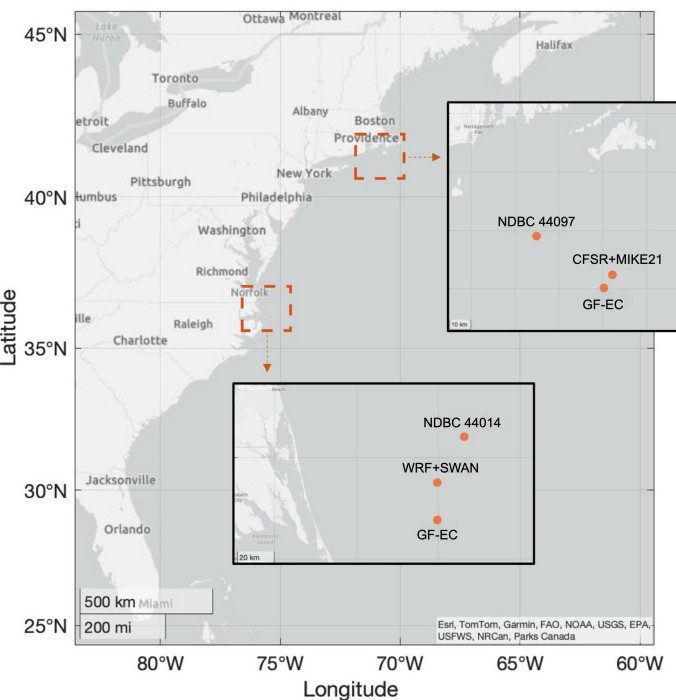

**Figure 1.** Location of study data: buoy observations and associated turbine analysis locations in the North Atlantic (top insert) and in the Mid-Atlantic Bight (bottom insert).

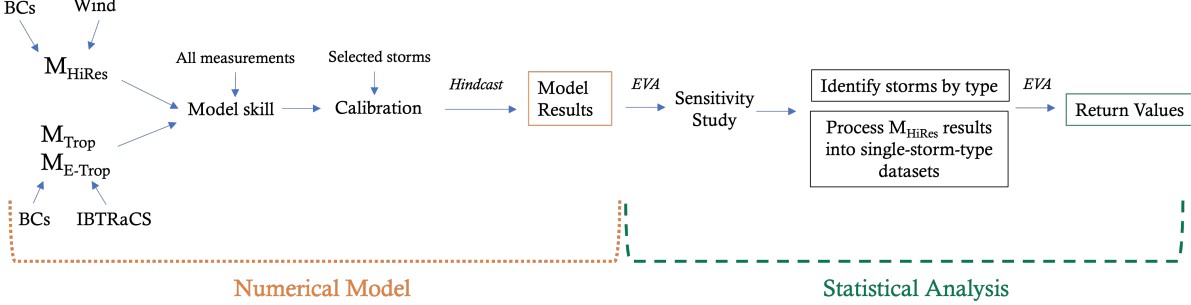

**Figure 2.** Study overview: Model calibration is applied linearly based on a subset of measurements, prior to statistical analysis by extreme value methods. Additional data sets are post-processed from the high-resolution models into time series of normal and single-type events. This process is applied in the North Atlantic and the Mid-Atlantic Bight analyses locations. Boundary conditions (BCs) and model parameterization can be found in Tables 1 and 2.



## 2.1 Model descriptions

Different spectral wave and hydrodynamics coupled models were chosen for the two investigated sites (Figure 1) in this study. These combinations, described in Tables 1 and 2, were chosen as representative modeling choices for quantifying the waves and hydrodynamics of the coastal United States. Both North Atlantic and Mid-Atlantic coupled models are locally validated.

| Model | Source | Resolution | Boundary Conditions | Duration |
|---|---|---|---|---|
| NA HiRes | CFSR | 22km spatial (2D) 2-hour | CFSR | 1979 - 2021 |
| MAB HiRes | WRF | 9km spatial (3D) 1-hour | CFSR | 1989 - 2019 |
| GROW-Fine East Coast | Tropical Boundary Layer Model | 5.5km spatial (3D) 15-minute | *Wind*: Satellite reconstruction *Pressure*: Far-field tropical reconstruction | *Trop*: 1924 - 2021 *Extra-Trop*: 1954 - 2021 |

**Table 1.** Wind parameterization for the three investigated models in the North Atlantic, Mid-Atlantic Bight, and along the US Atlantic coast (GROW-Fine East Coast).

| Model | Tool | Resolution | Boundary Conditions | Coupling | Spectral Parameterization |
|---|---|---|---|---|---|
| NA HiRes | MIKE21 | 600m wave (2D) 600m hydro (2D) 1-hour | DHI Global Waves (waves) DHI East Coast (hydro) | 1-way, hydro to waves | 36 directions 32 freq. bins 0.033 Hz min, freq. |
| MAB HiRes | SWAN+DELFT3D | 400m wave (2D) 400m hydro (3D) 1-hour | ERA5 (waves) HYCOM (hydro) | 2-way, waves and hydro | 36 directions 24 freq. bins 0.005 Hz min, freq. |
| GROW-Fine East Coast | OWI3G+ADCIRC | 5.5km wave (2D) 5.5km hydro (2D) 15-minute | GROW2012 (waves) Prevost '08 (hydro) | No dynamic coupling. Reanalysis of each modeled storm. | 48 directions 26 freq. bins 0.0029 Hz min, freq. |

**Table 2.** Wave and hydrodynamic parameterization for the three investigated models in the North Atlantic, Mid-Atlantic Bight, and along the US Atlantic coast (GROW-Fine East Coast).

A list of significant storms during the hindcast period, available observations, and storm events used for model calibration is provided in Appendix B.

### 2.1.1 North Atlantic high-resolution model

The North Atlantic model comprises 42 years of CFSR winds forcing a MIKE21 spectral wave and MIKE21 hydrodynamics hindcast, with one-way coupling from hydrodynamics to waves. The CFSR dataset is reanalyzed from the National Centers





for Environmental Prediction (NCEP) Climate Forecast System model. In this study, 2-hour averaged winds are interpolated to the hydrodynamic and wave domain resolutions, with a wind-wave coupling time step of 1 hour. The MIKE21 wave model is based on the wave action conservation formulation and is run in fully-spectral mode for this study. Depth-induced wave breaking ($\gamma = 0.8$ and $\alpha = 1$), spatially-varying whitecapping ($C_{ds} = 2.1$ to $2.6$), and nonlinear growth (coefficient $= 1.35$) are modeled. Wave calibration is based on a mixed-type set of 16 storms at four buoys, and is applied spatially throughout

the domain. A list of tropical storms during the hindcast period and nearby observations used for validation are provided in Appendix C.

Tides are modeled by the DTU10 Global Tide Model to capture any tidally-induced hydrodynamic or wave effects. More details on validation and the project background can be found in the Commonwealth Wind metocean report, Wrenger (2022).

This study investigates extremes at NA model location 40.75°N, 70.74°W, which has a depth of 62 m and is 45 km from

buoy 44097, the closest observation to the most "at-risk" turbine location. GROW-Fine East Coast ("GF-EC") model results are presented for 40.8°N, 70.7°W.

### 2.1.2 Mid-Atlantic model

The Mid-Atlantic model comprises 30 years of Weather Research and Forecasting (WRF) winds forcing a SWAN spectral wave and Delft3D hydrodynamics hindcast, with two-way coupling between the hydrodynamic and wave models. The WRF

model is a mesoscale atmospheric model developed by the National Center for Atmospheric Research (NCAR) and partners. In this study, vertically-nested domains with real lateral boundary conditions are applied and model the boundary layer with the YSU scheme. SWAN is a phase-averaged spectral wave model. Depth-induced wave breaking, whitecapping (Westhuysen formulation), and nonlinear growth are similarly modeled as for the North Atlantic case.

Tides are modeled by the Oregon State University TPXO dataset to capture any tidally-induced hydrodynamic or wave

effects. Wave calibration is based on observations of the 2020 "Hurricane Isaias" at 36.41°N, 75.23°W, and is applied uniformly throughout the domain. More details on validation and the project background can be found in the Kitty Hawk Wind metocean report, Georgas (2023).

This study investigates extremes at MAB model location 36.38°N, 75.00°W, which has a depth of 40 m and is 53 km from buoy 44014, the closest observation to the most "at-risk" turbine. GROW-Fine EC model results are presented for 36.2°N,

75.0°W.

### 2.1.3 GROW-Fine East Coast model

The GROW-Fine East Coast Tropical and Extra-Tropical models reconstruct storm winds and pressure fields based on multiple types of historical observations, including from satellites, aircraft flights, and SF microwave radiometetry. Waves are modeled by OWI3G, a wave model based on WAM that does not require prescribed wave spectrum for initialization, and are modified

by assimilation of ocean observations during storm events. The GF-EC domain spans from southern Florida through the Bay of Fundy.



To feasibly reconstruct 100 years of tropical storms and 75 years of extra-tropical storms, normal sea states are omitted from the dataset, and only major storm events are represented. The duration (associated data of storm development and decay) of individual events therefore varies for each storm throughout the dataset. The trajectory of tropical storms in the GF-EC Tropical
model is based on International Best Track Archive for Climate Stewardship (IBTRaCS) records.

To assess the influence of time (number of storms) on the statistical assessment of extremes, Extreme Value Analysis (EVA) is conducted for both the full duration of the GF-EC records and for the shorter duration of the high-resolution models: 1979–2020 (42 years), for the North Atlantic, and 1989–2019 (30 years) for the Mid-Atlantic Bight.

Model validation with NDBC buoy observations spanning the Atlantic coast from 1979, the beginning of data availability,
has been carried out by Oceanweather. For more details on the storm wind reconstruction method, boundary conditions, and overall model validation, refer to the GROW-Fine East Coast project description (Oceanweather (2022)).

### 2.1.4 Model skill

The performance of the North Atlantic model is assessed at 40.86°N, 70.808°W, which is 29 km from the nearest observation, NDBC 44097 from 2009 - 2020 (refer to Appendix C for Figure C1a, significant wave height, and Figure C1b, peak period).
Nine significant tropical and twenty significant extra-tropical events occurred during this period. There is good agreement in low-to-mid energy waves, however, model representation of the largest waves is higher than those recorded at NDBC 44097, resulting in an over-representation of significant wave height in this region, post-calibration, compared to buoy observations during the same time, suggesting that modeled sea states over the hindcast period were larger than observed for the strongest events. However, when model and observation quantiles are compared for tropical-cyclone-only events, the model root mean
square error (RMSE) increases from 0.338 to 0.474, suggesting poorer performance during these events.

Model skill in the Mid-Atlantic model is assessed at 36.38°N, 75.00°W, which is 29 km from the closest observation with the longest overlapping record, NDBC 44014 from 1990–2019 (refer to Appendix C for Figure C1c, significant wave height, and Figure C1d, peak period). Twenty-seven significant tropical and thirty-five significant extra-tropical events occurred during this period. The model shows good agreement in low-to-mid energy waves, however, under-representation is observed for large-
amplitude and large-peak-period swell, suggesting that the model may not suitably capture sea states during tropical cyclone events. When model and observation quantiles are compared for tropical-cyclone-only events, the RMSE is slightly raised from 0.455 to 0.468, suggesting weaker model performance during these events, as well.

## 2.2 Univariate extreme value methods

Return values are determined according to the general linear model,

$$Y = -\beta' \mathbf{x} + \epsilon \tag{1}$$

which associates covariates $x$ to the return values $Y$ in terms of regression coefficient, $\beta'$ and error, $\epsilon$. The return values are calculated based on the log-likelihood of the density function, described here as either block maxima (the Gumbel form of the





Generalised Extreme Value distribution, GEV) or as probability-based peaks-over-threshold (Generalized Pareto, GP), when assessing the sensitivity of results to extreme value method choice.

### 2.2.1 Block Maxima


The Generalized Extreme Value distribution describes a set of data in terms of $\xi$, shape, $\mu$, location, and $\sigma$, scale. When $\xi = 0$, this distribution is equivalent to the 2-parameter Gumbel cumulative distribution function used in this study:

$$G(x) = exp\left(-exp\left(-\frac{x-\mu}{\sigma}\right)\right), \quad for \quad -\infty < x < \infty \tag{2}$$

from Coles (2001). The location and scale variables were determined by empirical estimation, and are provided in Appendix C. The associated return value, $x_p$, for probability period $p$ are calculated as:


$$x_p = \mu + \sigma\left(-ln\left(-ln(1-p)\right)\right), \quad for \quad -\infty < x_p < \infty \tag{3}$$

The statistical basis for the distribution fit is composed of the annual largest values of the model data.

### 2.2.2 Separating storm types

The high-resolution hindcasts comprise all normal and mixed-type storm periods in one dataset. In order to calculate extreme return values according to storm type, as established by Gomes and Vickery (1978), the high-resolution model results were post-processed for their entire hindcast period into two datasets: one, with tropical storm events removed, and another with extra-tropical storm events removed, aligned with storms represented in the overlapping period with the GROW-Fine datasets. A list of tropical cyclones included in these datasets is available in Appendix C.


Extreme Value Theory assumes that extremes are independent variables (Mackay and Johanning (2018)). However, multiple peaks may be attributed to the same event during storm growth and dissipation. To preserve the independence criterion in this study, only the peak significant wave height is retained in a period of 98 hours (storm length also assumed by Kresning et al. (2024), Oceanweather (2022), and others) during an identified storm. For simplicity, the same storm duration is assumed for both tropical and extra-tropical events, however in a number of instances, the storm duration is shorter than this period. In general, tropical cyclone forward speed is highly dependent on local climatic conditions, which influences storm duration in a given location. For storm removal from a dataset, values 49 hours prior to and 49 hours after the peak significant wave height are removed.



## 3 Results and Discussion

As return periods extend to 50 years and beyond, tropical cyclones are observed to be the dominating storm type for extremes in both the North Atlantic (Figure 3a), and for all return periods in the Mid-Atlantic (Figure 3b). While extra-tropical cyclones are a more frequent occurrence at the North Atlantic site (there are 48 significant extra-tropical cyclone events recorded during




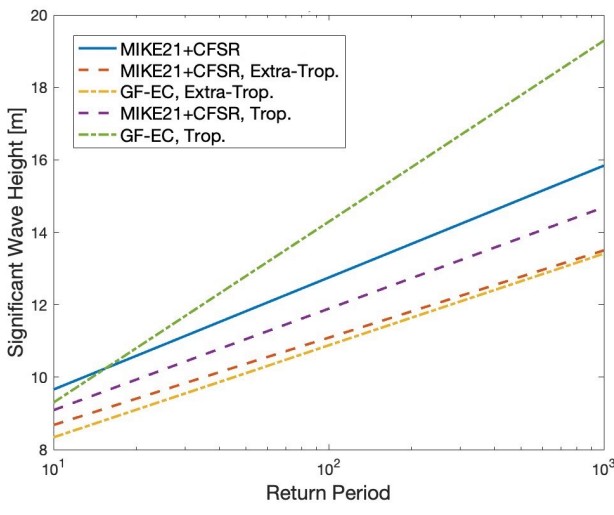

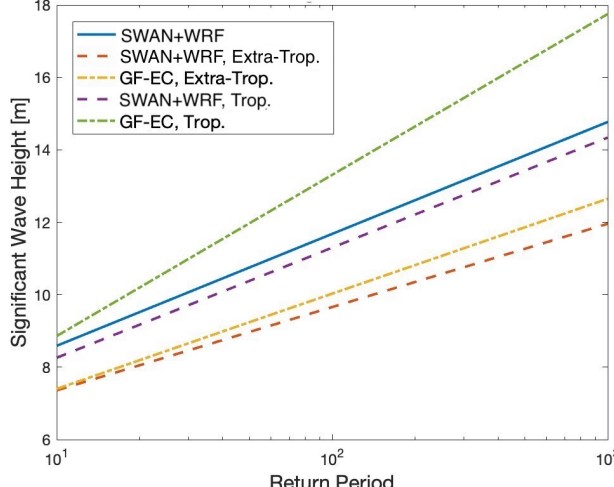

(a) Significant wave height return values for the North Atlantic site at 40.8N, 70.7W.

(b) Significant wave height return values in the Mid-Atlantic Bight site at 36.2N, 75.0W.

**Figure 3.** Study return values for all model results by annual maxima with Gumbel distribution.

the hindcast period, whereas there are 15 significant tropical cyclone events), tropical cyclones were able to reach greater intensities in the studied record.

The return values calculated from the GF-EC Tropical model show diverging values, in trend and magnitude, from the high-resolution hindcast for both the North Atlantic (Figure 3a, dash-dotted green line) and for the Mid-Atlantic (Figure 3b, dash-dotted green line) scenarios, despite different calibrations, boundary conditions, and latitudes. The influence of statistical and modeling choices on these diverging trends for tropical cyclone events are investigated further in the following sections.

In contrast, there is good agreement between the GF-EC Extra-Tropical model and the high-resolution model data, in both its original and post-processed (normal + extra-tropical events) data set forms (for the North Atlantic, see Figure 3a, dash-dotted orange line, and for the the Mid-Atlantic Bight, see Figure 3b, dash-dotted orange line). This suggests that both the storm physics and storm sample set of extra-tropical events at these latitudes are sufficiently represented by the high-resolution models in the 30–42 year periods investigated.

### 3.1 Influence of Model Characteristics

#### 3.1.1 Overall model performance

Representative model performance is presented here for similar-intensity storms: the January 2015 extra-tropical storm (Class 3 on the Dolan-Davis scale) and "Hurricane Dorian" (Category 2, Saffir-Simpson scale). The time and duration of the extra-tropical wave height successive peaks (Figure 4a) agree between models and measurements, although the GF-EC Extra-Tropical model over-represents the largest peak as compared to measurements. Evolution of total wave period was captured



very well for the extra-tropical event as exhibited by overlapping timeseries evolution of both model results and buoy obser-
vations (Figure 4b). While there is overall agreement between models and measurements on peak timing and storm duration
during "Hurricane Dorian", the the Mid-Atlantic Bight high-resolution model shows under-representation of the storm peak
significant wave height (Figure 4c) by 10 % as compared to observations, and by 15 % as compared to the GF-EC tropical
model. While the observed storm peak period (Figure 4d, green crosses) spans from 4 to 13 seconds, neither model captures
the higher-frequency, wind-driven waves below 7 seconds in this event. The MAB high-resolution model follows the largest
periods, while the the GF-EC Tropical model captures frequencies in the middle of the range.

Throughout the 30-year period of the MAB high-resolution hindcast, all identified tropical and extra-tropical events are
represented. There is a notable trend of under-representation of the largest storm peaks, suggesting that calibration based on a
single tropical cyclone observation was insufficient in terms of numerical modeling. However, calibration is not the only (or
largest) influence on the return values calculated in this study, as discussed in Section 3.2.

Similarly, the North Atlantic model represents all significant tropical and extra-tropical events on record during the 42-year
period, in a number of cases with higher significant wave heights than in the GF-EC Tropical dataset. Minor differences between
the models are investigated in the following sections. There is one case of substantial under-representation of significant wave
height, however: during "Hurricane Bob", when the analysis point was within the storm fetch. Compared to the GF-EC Tropical
model for this event, this was a consequential error for a dataset containing 15 tropical cyclone events. The maximum significant
wave height in the GF-EC Tropical dataset during "Hurricane Bob" is on the order of the two largest wave heights in the high-
resolution data set: "Hurricane Gloria" of 1985, and the "Storm of the Century" Blizzard of 1993. Given the rarity of this wave
magnitude for this location, this numerical modeling error is expected to influence the overall statistical assessment of extreme
values for the site.

### 3.1.2 Tropical Cyclone Sea State Representation

Model performance is first assessed for the representation of wind-sea- and swell-waves generated during storm evolution.
For brevity, only the North Atlantic case is presented here. The two-dimensional wave spectrum is partitioned (separated into
wind-sea and swell wave systems) by the watershed algorithm. (See Portilla-Yandun et al. (2009) for more information on
spectral partitioning schemes.) For both the high-resolution and GF-EC data sets, the wave components are similarly captured
during "Hurricane Bob" and during "Hurricane Dorian". Both models capture the physical progression of a tropical cyclone as
signaled by the evolution of wave peak period: elevated swell frequencies (approximately 10 seconds, Bob, GF-EC Tropical
in Figure 5a and NA high-resolution in Figure 5b; up to 16 seconds, Dorian, GF-EC Tropical in Figure 5c and NA high-
resolution in Figure 5d) precede maximum winds. In the case of "Hurricane Bob", where the analysis point is within the storm
fetch, peak significant wave height (solid vertical line) corresponds to the wind-sea and overall storm maximum peak periods
(approximately 15 seconds). Both swell and wind-sea frequencies are characterized similarly by both the high-resolution and
GF-EC Tropical models.

In the case of "Hurricane Dorian", where the analysis point is outside of the storm fetch, the swell maximum peak period (16
seconds) and wind-sea maximum peak period in both models arrive prior to the time of peak significant wave height (vertical

(a) Hs during the 2015 Blizzard, North Atlantic

(b) Tp during the 2015 Blizzard, North Atlantic

(c) Hs during "Hurricane Dorian", Mid-Atlantic Bight

(d) Tp during "Hurricane Dorian", Mid-Atlantic Bight

**Figure 4.** Representative performance of the high-resolution and GF-EC model ocean wave results during (a–b) extra-tropical and (c–d) tropical cyclone storm events, with buoy measurements. To protect intellectual property and to preserve scale, significant wave height values are normalized to the single largest peak value.



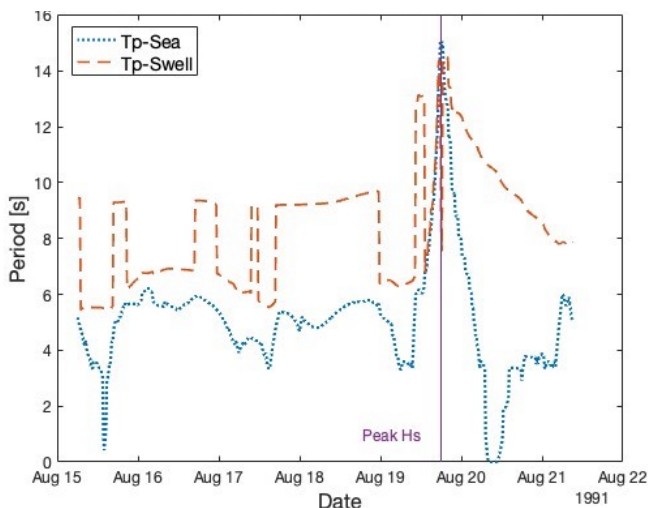

(a) Elevated sea state during "Hurricane Bob", GF-EC Tropical model

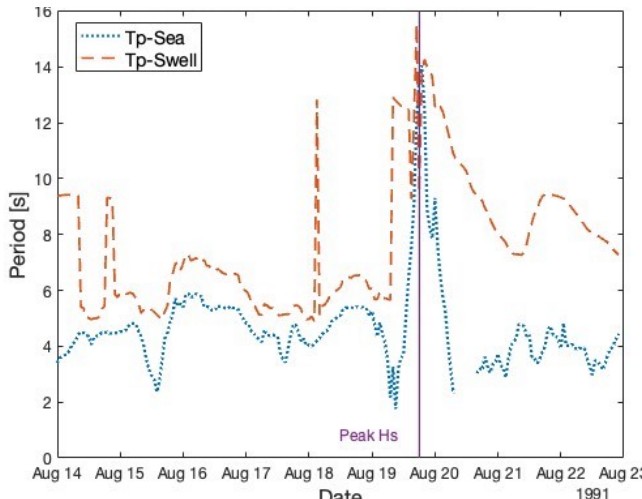

(b) Elevated sea state during "Hurricane Bob", NA high-resolution model

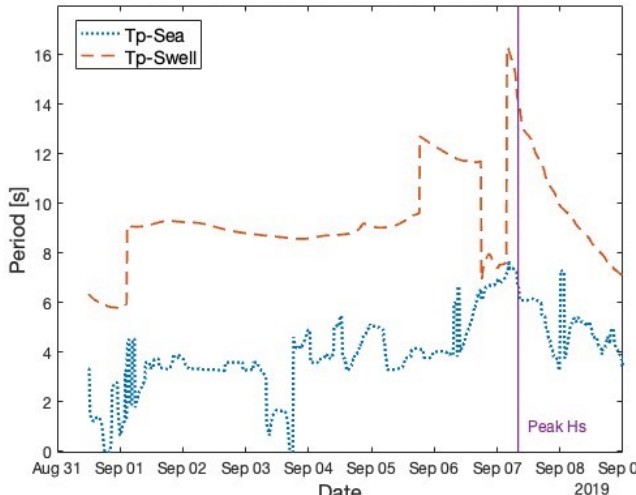

(c) Elevated sea state during "Hurricane Dorian", GF-EC Tropical model

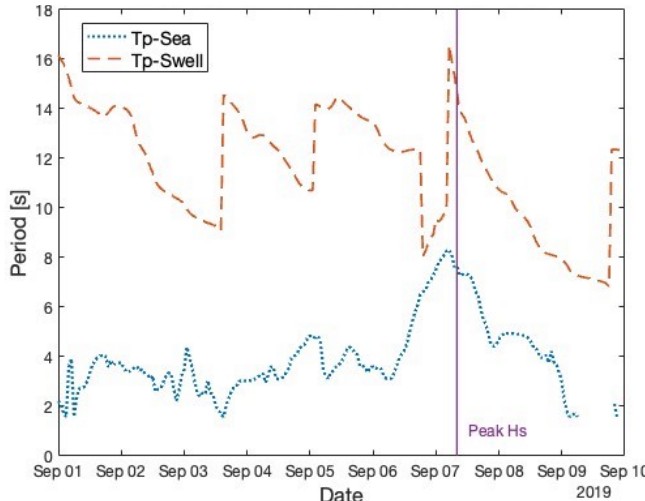

(d) Elevated sea state during "Hurricane Dorian", NA high-resolution model

**Figure 5.** Partitioned wave systems during the arrival and passage of two tropical cyclone events by the high-resolution and GF-EC Tropical models for the North Atlantic site.

solid line). During and after this time, both models represent "Hurricane Dorian" with similar magnitudes of wind sea and swell, however, there is a notable difference in swell frequencies prior to the storm peak between the two models. Section 3.1.4 investigates this discrepancy further.





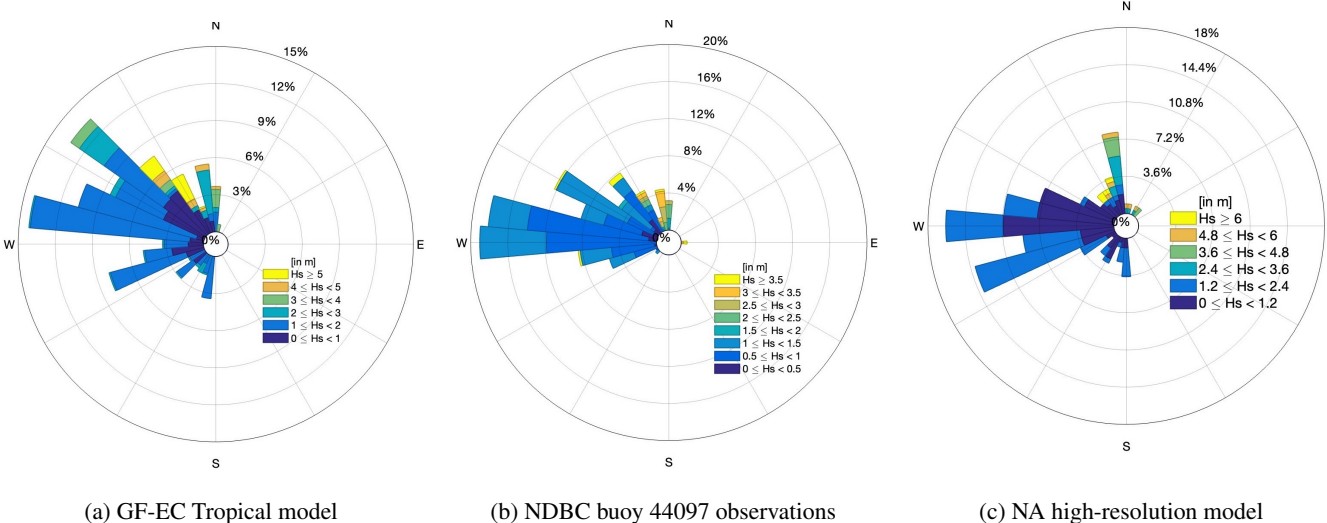

(a) GF-EC Tropical model      (b) NDBC buoy 44097 observations      (c) NA high-resolution model

**Figure 6.** Mean wave direction during passage of "Hurricane Dorian" at the North Atlantic site.

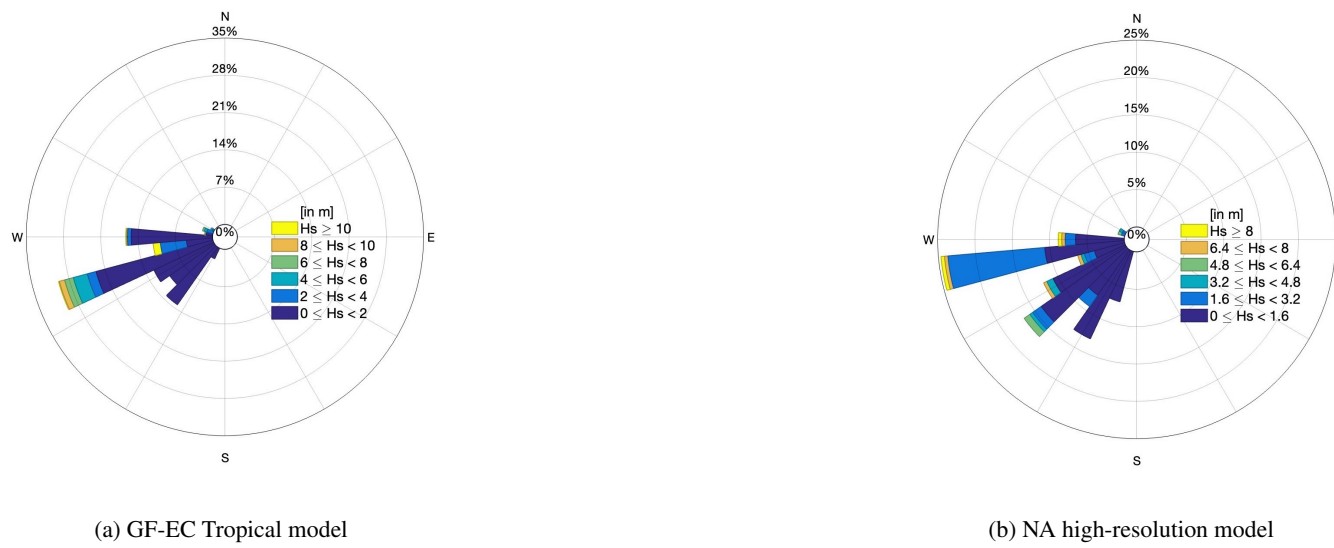

(a) GF-EC Tropical model                          (b) NA high-resolution model

**Figure 7.** Mean wave direction during passage of "Hurricane Bob" at the North Atlantic site.

### 245 3.1.3 Wave directional spreading

In the case of "Hurricane Dorian", the high-resolution and GF-EC models show global similarities in mean wave direction, however, the NA high-resolution model (which represents 36 directional sectors and 32 frequency bins), Figure 6c, presents a narrower range of the largest waves from the North-North West as compared to GF-EC Tropical (which simulates 48 directional sectors and 26 frequency bins), Figure 6a, and buoy observations, Figure 6b.





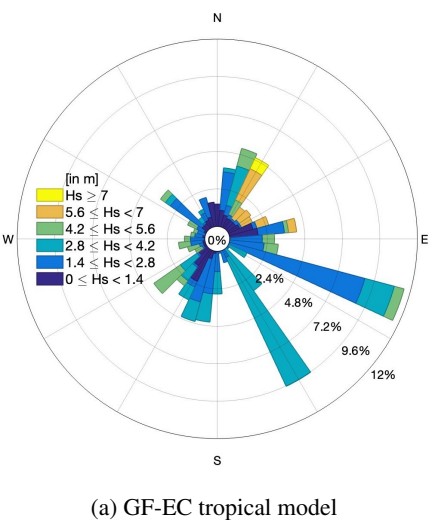

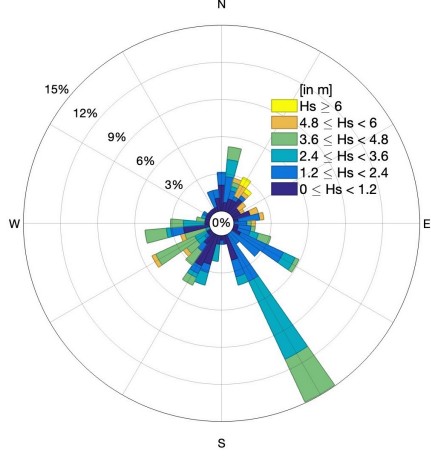

(a) GF-EC tropical model

(b) NA high-resolution model

**Figure 8.** Mean wave direction during passage of the 2015 Blizzard at the North Atlantic site.

In the case of "Hurricane Bob" in the North Atlantic, where analysis occurs inside the storm fetch, both models represent a concentrated range of waves similarly for the average and largest waves (GF-EC, Figure 7a, and high-resolution, Figure 7b), while the magnitude of the wave height differs. This narrower range agrees with Forristall and Ewans (1998), which confirms that spreading factors are generally reduced the closer they are to the storm center. Storm winds (90 knots, or 46 m/s) were recorded in this location and passage of the storm eye over the area took one hour. Considering wave directions represented,

the GF-EC Tropical model captured seas on a 15-minute interval. Wave buoy measurements occurred on a 30-minute cycle, and the NA high-resolution model modeled seas on an hourly time step. Given the high-intensity, fast-moving event through the region, an hourly time step likely contributed to the under-representation of sea states in the high-resolution model during "Hurricane Bob".

     In contrast, a large directional spread is observed in both models during the 2015 Blizzard (Figure 8a, GF-EC Extra-Tropical,

and Figure 8b, NA high-resolution). This again agrees with the findings of Forristall and Ewans (1998) that, while ocean response to tropical cyclones may result in a variety of wind sea and swell wave system interactions, there is a generally narrower range of directional spreading for tropical cyclones than for extra-tropical cyclones. Given the larger spread of wave directional propagation expected during extra-tropical storms, combined with excellent alignment in magnitude and trend with buoy observations of significant wave heights, the high-resolution models are considered highly capable of resolving the

physical properties of wave development by extra-tropical events on the Atlantic coast.

### 3.1.4 Proximity to Tropical Cyclone Eye

The physical evolution of tropical cyclones varies from to storm to storm. The radius of a tropical cyclones can span from 20 to 250 km, while the radius of an extra-tropical cyclone can span 100 km to 2000 km. The representation of waves and wave



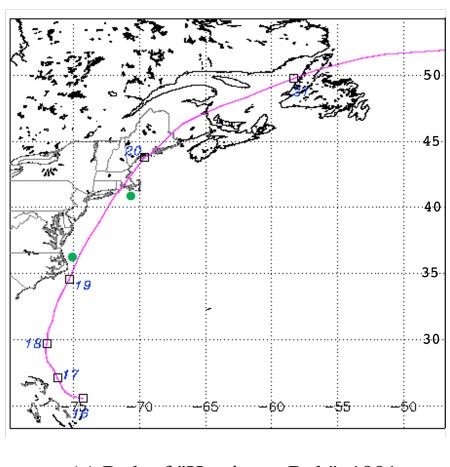

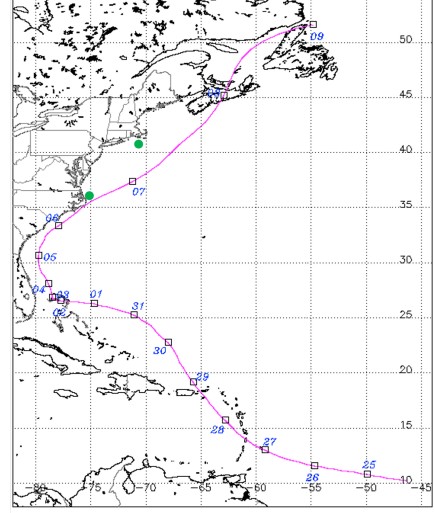

(a) Path of "Hurricane Bob", 1991        (b) Path of "Hurricane Dorian", 2019

**Figure 9.** Historical best tracks of the studied tropical cyclones, from www.csics.org/ibtracs

growth due to a tropical cyclone in design values may therefore be sensitive to storm track, fetch, and the location of derived
site conditions. The two tropical cyclone events investigated in detail here represent a close proximity to the storm path: From
IBTRaCS records, the eye of "Hurricane Bob" reached its peak low pressure (950 mb, Category 3) 60km from the Mid-Atlantic
analysis point, and came within 100km of the North Atlantic analysis point on August 19, 1991 as a Category 2 storm. The
eye of "Hurricane Dorian" also came within 60km of the Mid-Atlantic analysis point as a Category 2 storm, and persisted as
Category 2 within 260km from the North Atlantic analysis point.

As Hwang and Walsh (2018) show based on satellite observations of multiple tropical cyclone events, wave growth inside
tropical cyclones follows the same formulation as wave growth in other, non-tropical-cyclone conditions. As a result, Hwang
and Walsh (2018) describe fetch and duration as linearly relating to storm radius. Differences in wave development in this
study are therefore considered primarily to be a function of fetch or duration representation, and not model performance.

To further quantify the influence of storm fetch on peak significant wave heights, the momentum flux between the atmosphere
and the ocean surface is investigated in terms of wind stress (from Jones (2011)):

$$\tau_i = \rho C_D U_{10m,i}^2 \tag{4}$$

where $\rho$ is air density and $C_D$ is the coefficient of drag. Setting aside constants, effective stress is a function of the square of
the wind speed, normalized here to the largest magnitude recorded from either the GF-EC Tropical and NA high-resolution
models during the event. The square of normalized wind speed inside of the storm radius for "Hurricane Bob" is presented in
Figure 10a, and outside of the storm radius for "Hurricane Dorian" in Figure 10b. There is a notable difference in these cases
in applied wind forcing to the sea surface, despite both storms persisting as Category 2 at the time and location of analysis.





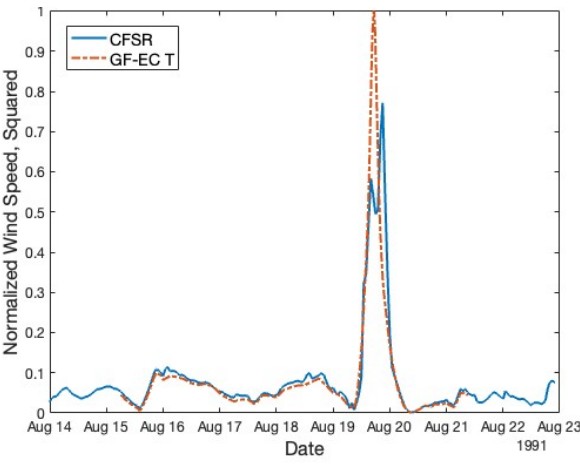
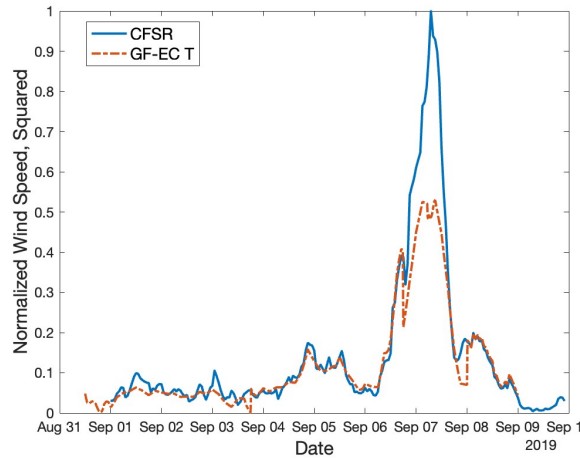

(a) GF-EC Tropical and NA high-resolution normalized wind, squared, within the storm radius (fetch) of "Hurricane Bob".

(b) GF-EC Tropical and NA high-resolution normalized wind, squared, outside of the storm radius (fetch) of "Hurricane Dorian"

**Figure 10.** Effective normalized wind stress on ocean surface during two tropical cyclones at the North Atlantic site.

During the modeled events of "Hurricane Bob", the NA high-resolution model applies 25 % less wind stress on the ocean surface than the GF-EC tropical simulation; peak significant wave height from the coupled MIKE21 wave model is 25 % lower than in the assimilated GF-EC Tropical wave dataset. In contrast, during "Hurricane Dorian", the NA high-resolution peak storm winds delivered 48 % larger ocean stress; the assimilated waves in the GF-EC Tropical dataset were 10 % lower than in the high-resolution model. Drawing from the discrepancies between peak winds in the two models, relying on calibration to compensate for under-representation by the model of wave heights may not be sufficient without methodical selection of observations from within and from outside of the tropical cyclone fetch to calibrate against.

As stress on the ocean surface is a function of the square of wind velocity (equation 4), the height of non-fully-developed wind waves generated by tropical cyclones are sensitive to the square of the error in peak modeled wind speed. Under-representations of these values, as noted in Campos et al. (2022) and Caires and Sterl (2005), in conjunction with characterization of points solely outside of the storm fetch, can lead to the under-representation of modeled wave growth and significant wave heights.

### 3.2 Influence of Statistical Choices

In the North Atlantic, where extra-tropical cyclones are an annual event, the 10-year return value for the high-resolution model shows a characteristic of mixed species analysis: the influence of these events lifts the overall return value over the tropical-cyclone-only values (see the intersection of the "CFSR+MIKE21" and "GF-EC Trop. Full" trend lines after the 10-year mark, Figure 11a). However, the opposite is observed at longer return periods: the influence of the less-intense storm type appears to



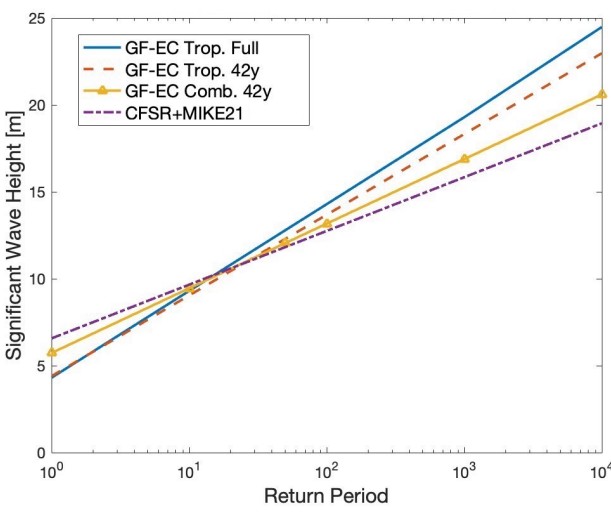

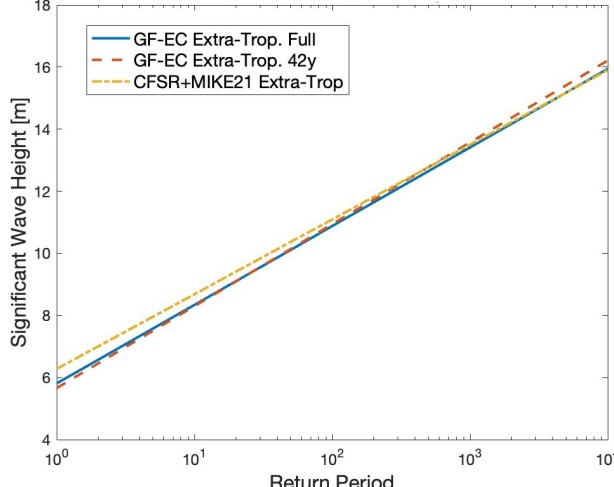

(a) Tropical-only return values from the GF-EC Tropical, high-resolution, and post-processed GF-EC model data sets.

(b) Extra-Tropical-only return values from the GF-EC Extra-Tropical, high-resolution, and post-processed GF-EC model data sets.

**Figure 11.** Return values for single-storm conditions at the North Atlantic site from the high-resolution and post-processed GF-EC data sets.

reduce the overall return value, leading to increasing divergence between the "GF-EC Trop. Full" and "CFSR+MIKE21" trend
lines.

When assessing return value sensitivity to dataset length, or the number of extreme events represented in the data, the GF-EC Tropical and Extra-Tropical datasets are shortened to 42 years (for the North Atlantic) and 30 years (for the Mid-Atlantic Bight). The 10- to 10,000-year return values reduced slightly in the North Atlantic case (Figure 11a) and increased slightly in the Mid-Atlantic case (Figure 12a). In neither scenario does this shortened dataset explain the trend and magnitude differences
between the two sets of results. On the contrary, the increase in return values observed in the Mid-Atlantic case is influenced by higher-than-average tropical cyclone activity in the 1990's and 2000's, and misses the lower-than-average period of the 1970's and 1980's.

Similarly, analysis of the post-processed high-resolution data into a single storm type data did not mitigate differences between the GF-EC Tropical- and high-resolution-derived return values in Figures 11a and 12a; in fact, these values are further
reduced from the original high-resolution data set. In both locations, these lower values suggest that the periods of 1989–2019 (Mid-Atlantic) and 1979–2020 (North Atlantic) do not present a sufficient basis for fully characterizing extremes due to tropical cyclones. As previously mentioned, the under-representation in the high-resolution model of one of the largest peaks in the dataset, "Hurricane Bob", also likely contributes to the lower projected return values in the North Atlantic case. To mitigate this effect due to limited dataset duration of the high-resolution models, grid point "pooling" proposed by Heideman and Mitchell
(2009) may be worthwhile.





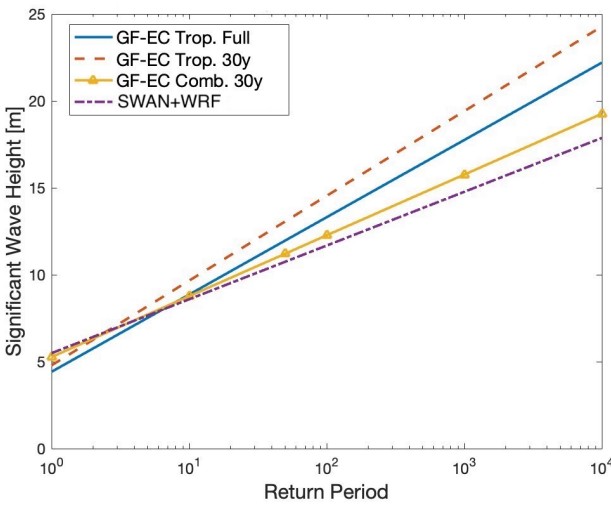
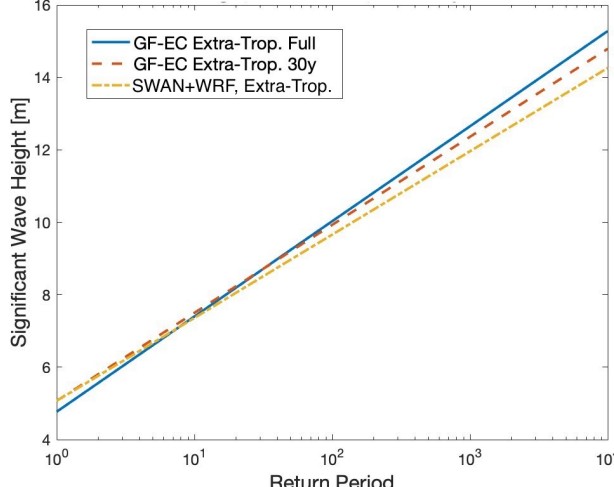

(a) Tropical-only return values from the GF-EC Tropical, high-resolution, and post-processed GF-EC model data sets.

(b) Extra-Tropical-only return values from the GF-EC Extra-Tropical, high-resolution, and post-processed GF-EC model data sets.

**Figure 12.** Return values for single-storm conditions at the Mid-Atlantic Bight site from the high-resolution and post-processed GF-EC data sets.

This trend is not observed in either location for extreme values due to extra-tropical cyclones (Figure 11b, North Atlantic, and Figure 12b, Mid-Atlantic). Differences of 1.2 %–1.5 % were observed in the 1,000- and 10,000-year cases (respectively) for the North Atlantic, and of 2.3 %–3.3 % for the Mid-Atlantic Bight. The numerical and statistical representation of extra-tropical extremes by the high-resolution model duration, configuration, and boundary conditions appears sufficient in both the

North Atlantic and Mid-Atlantic cases.

When the GF-EC Tropical and GF-EC Extra-Tropical model results are combined into a single dataset and analyzed together in one distribution, return values reduce to values close to those derived from the high-resolution models (see Figure 11a, GF-EC "Combined" 42-year dataset, and Figure 12a, GF-EC "Combined" 30-year dataset). The cumulative effect of a reduced storm sample (shortening the GF-EC Tropical data from 100 to 42 and 30 years) and single statistical analysis the mixed storm

types explains much of the differences between return values derived from the high-resolution model and from the GROW-Fine tropical models in Figures 3a and 3b: extreme values calculated from this "Combined" dataset are up to 15.9 % lower than the Tropical-only case in the North Atlantic and up to 13.2 % lower than the Tropical-only case in the Mid-Atlantic. These differences occur similarly in both locations, despite differences in tropical storm frequency and peak storm intensities observed in the historical record.





## 4 Conclusions

In this study, the factors influencing extreme significant wave height estimation due to tropical cyclones were assessed for the relative influence of model and statistical choices at two locations on the US Atlantic coast. The performance of two high-resolution models with differing calibration methods were assessed alongside reanalyzed ocean wave values, forced by reconstructed storm winds, during a period spanning from 30 to 100 years. Choices leading to the statistical distribution of extreme events with the block maxima method were also investigated.

Overall, return values due to extra-tropical events are shown in this study to be well-resolved by established methods of metocean modeling with CFSR- or WRF-generated winds and ERA5-boundary conditions, as shown by the model performance and directional wave spreading compared to observations and by similar return values generated from all models investigated. However, the differences in extreme values calculated in this study when only considering tropical cyclones suggests that under-representation of peak wave parameters by the high-resolution models can not be mitigated by calibration alone, due in part to the model temporal resolution of wind-generated waves, and in part to the limited number of observations of storms from a range of locations within and outside of the storm fetch. While both high-resolution models, with different wind forcing, model design, boundary conditions, and calibration techniques captured tropical cyclone peak significant wave heights within 7 % on average for both the NA and MAB high-resolution time series, there were larger differences for the largest storm waves.

In addition to numerical representation, statistical choices made during extreme value analysis were a major factor that contributed to the difference between calculated return values. The sensitivity of values derived from the GF-EC Tropical datasets when reduced to the high-resolution time periods, in addition to the reduction of values when the high-resolution datasets were post-processed to single-storm time series suggests that further assessment of the number of minimum sufficient number of storms is required for the proper characterization of a site, and that this dataset length requirement differs between tropical and extra-tropical assessment, due to observed variations in storm size and intensity. The subsequent calculation of extreme values based on a single distribution of these two storm types over a smaller number of storm events resulted in the under-estimation of 100-, 1,000- and 10,000-year design values in this study for both the North Atlantic (18 %, 1,000-year Hs) and Mid-Atlantic Bight (17 %, 1,000-year Hs) locations between the GROW-Fine East Coast and high-resolution models.

For more accurate determination of return values for offshore infrastructure design in areas with tropical cyclone activity, the following is recommended:

- Analysis of a point or range of points within the storm wind radius: proximity to the storm eye of 200km or less, depending on storm size.

- Extreme value analysis should be carried out on single-storm-type datasets.

- A 30- to 40-year hindcast period is sufficient to characterize extra-tropical extremes in the North Atlantic and Mid-Atlantic locations investigated.

- A 30- to 40-year hindcast period is not sufficient in the investigated areas to characterize tropical cyclone extremes. If a longer data period is not available, grid point pooling is necessary.



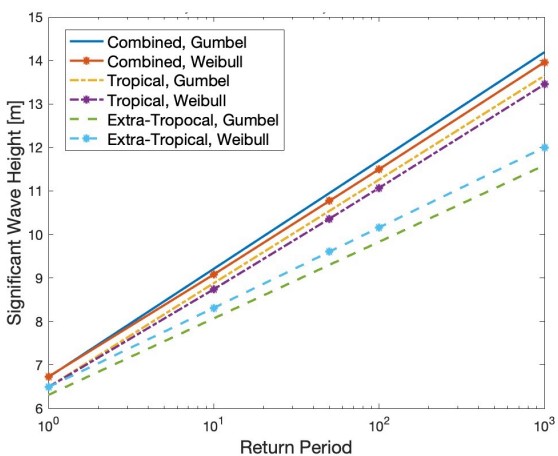
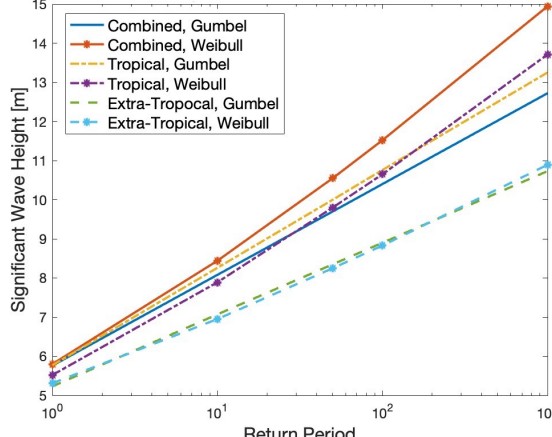

(a) Return values at the North Atlantic site from the high-resolution model results and two post-processed subsets of the high-resolution model results based on storm type.

(b) Return values at the Mid-Atlantic Bight site from the high-resolution model results and two post-processed subsets of the high-resolution model results based on storm type.

**Figure A1.** Return values for the high-resolution model results when fit with Gumbel and Weibull distributions.

*Data availability.* Original high-resolution wave model time series at 40.8°N, 70.7°W and 36.2°N, 75.0°W are available at https://doi.org/10.5281/zenodo.13884957. Due to intellectual property protection, the GROW-Fine EC data is not publicly available.

**Appendix A: Comparing Extreme Value Method Return Values**

The sensitivity of extreme values to the Gumbel distribution and block maxima method employed in this study was conducted for a subset of the high-resolution model results with the peaks-over-threshold method. The POT analysis sequence is based on a Generalized Pareto distribution cumulative distribution function, where $u$ describes the selected threshold, $\xi$ and $\sigma$ describe the shape and scale of the distribution, respectively, and and $\lambda$ is the number of annual exceedences, such that $x > u$:

$$1 - p = exp\left(-\lambda\left(1 + \xi\frac{x_p - u}{\sigma}\right)^{-1/\xi}\right), \quad for \quad \xi \neq 0 \tag{A1}$$

(from Jonathan and Ewans (2013)). The statistical basis is composed of values that exceed a physically realistic threshold, which can be subjective to user choice.

Minor differences due to data distribution fit are observed at the North Atlantic site and remain constant for longer return periods. Minor differences due to data distribution, which increase with return period, are observed at the Mid-Atlantic Bight
site, resulting in a maximum difference of 8 % in the 10,000-year case. Overall, the chosen fit is therefore not considered to influence the statistical trends investigated in this study.

When comparing values derived from the two extreme value methods, the largest discrepancy is observed between the post-processed "tropical" results. Peaks-over-threshold returns larger 10,000-year values than by block maxima in the Mid-Atlantic





Bight (10 % larger, POT-Weibull; 3 % larger, POT-Gumbel). In the North Atlantic, analysis by peaks-over-threshold returned
smaller 10,000-year values than by block maxima (9.3 % smaller, POT-Weibull; 8.2 % smaller, POT-Gumbel). Given that there
is no significant bias between the two locations of one method or the other, the block maxima method, which is less sensitive
to user choices, is considered suitable for the study.

## Appendix B:  Tabulated Extreme Values

| Model | 50 | 100 | 1000 | 10,000 | $\mu$ | $\sigma$ |
|---|---|---|---|---|---|---|
| NA high-res. (BM) | 11.82 | 12.75 | 15.82 | 18.94 | 1.34 | 6.57 |
| NA GF-EC Trop. (BM) | 12.79 | 14.29 | 19.29 | 24.48 | 2.17 | 4.31 |
| NA GF-EC Extra-Trop. (BM) | 10.11 | 10.88 | 13.42 | 15.94 | 1.10 | 5.86 |
| MAB high-res. (BM) | 10.00 | 10.76 | 13.25 | 15.74 | 1.34 | 5.49 |
| MAB GF-EC Trop. (BM) | 11.97 | 13.31 | 17.75 | 22.2 | 1.93 | 4.42 |
| MAB GF-EC Extra-Trop. (BM) | 9.24 | 10.03 | 12.65 | 15.28 | 1.51 | 4.65 |

**Table B1.** Selected return period extreme significant wave heights [m] and associated fit parameters for Block Maxima (BM) analysis at the
North Atlantic (NA) and at the Mid-Atlantic Bight (MAB) locations.

## Appendix C:  Tropical Cyclone Events During the Hindcast Period

The high-resolution model skill was assessed at the nearest buoy to turbine analysis locations in the North Atlantic and Mid-
Atlantic Bight project areas, presented as quantile-quantile plots in Figures C1a - C1d. Calibration of the North Atlantic
high-resolution model was conducted based on observations (Table C1) throughout the model domain. All tropical cyclone
events that occurred during the high-resolution forecast and buoy-based observations of these events are provided in Table C2.

| Buoy ID | Coordinates | Overlapping Period |
|---|---|---|
| NDBC 44008 | 40.496N, 69.250W | March 26, 2003 – December 31, 2020 |
| NDBC 44017 | 40.693N, 72.049W | April 30, 2008 – December 31, 2020 |
| NDBC 44020 | 41.497N, 70.283W | March 10, 2009 – December 31, 2020 |
| NDBC 44097 | 40.967N, 71.124W | September 17, 2009 – December 31, 2020 |

**Table C1.** Buoy observations from the National Data Buoy Center (NDBC) used for North Atlantic model calibration.




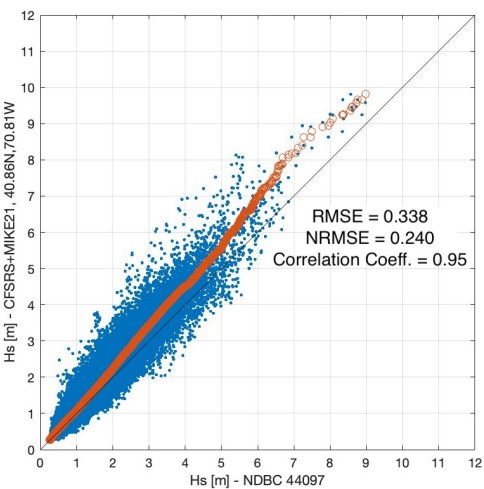

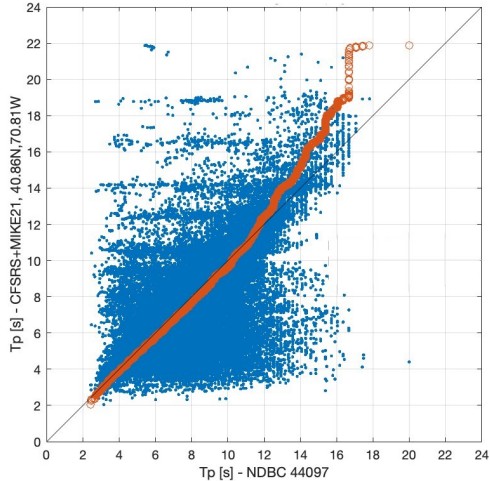

(a) NA high-resolution model skill for significant wave height (Hs) at 40.86°N,70.81°W, compared to concurrent NDBC buoy 44097 observations.

(b) NA high-resolution model skill for peak period (Tp) at 40.86°N,70.81°W, compared to concurrent NDBC buoy 44097 observations.

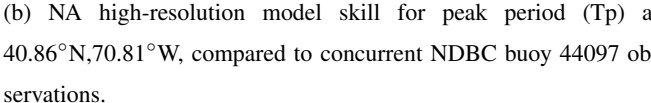

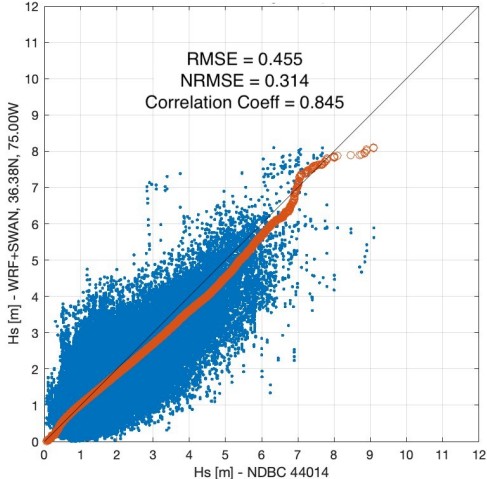

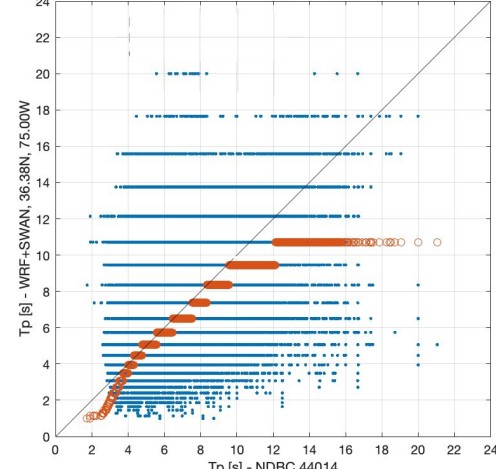

(c) MAB high-resolution model skill for significant wave height (Hs) at 36.38°N, 75.00°W, compared to concurrent NDBC buoy 44014 observations.

(d) MAB high-resolution model skill for peak period (Tp) at 36.38°N, 75.00°W, compared to concurrent NDBC buoy 44014 observations.

**Figure C1.** Quantile-Quantile comparison of calibrated high-resolution model wave results with measurements. Blue dots represent data in the overlapping time period, and red circles represent the quantiles in the associated regime. The black diagonal line represents a 1:1 correspondence between model and measurement data.





| Storm (Approx. day of peak) | North Atlantic Observations | Mid-Atlantic Bight Observations | In GF-EC NA model? | In GF-EC MAB model? |
|---|---|---|---|---|
| August 19, 1991 | | NDBC 44014 | X | X |
| October 30, 1991 | | NDBC 44014 | X | X |
| September 1, 1993 | | NDBC 44014 | | X |
| November 18, 1994 | | NDBC 44014 NDBC 44019 | | X |
| August 17, 1995 | | NDBC 44014 | | X |
| July 13, 1996 | | NDBC 44014 | X | X |
| September 6, 1996 | | NDBC 44014 | | X |
| October 8, 1996 | | NDBC 44014 | | X |
| August 28, 1998 | | NDBC 44014 | | X |
| September 1, 1999 | | NDBC 44014 | | X |
| September 16, 1999 | | NDBC 44014 | X | |
| October 18, 1999 | | NDBC 44014 | | X |
| September 10, 2002 | | NDBC 44014 | | X |
| September 28, 2003 | | NDBC 44014 | X | X |
| August 3, 2004 | | NDBC 44014 | X | X |
| October 25, 2005 | | NDBC 44014 | | X |
| November 3, 2007 | | NDBC 44014 NDBC 44056 | X | X |
| September 6, 2008 | | NDBC 44014 NDBC 44056 NDBC 44100 | | X |
| September 3, 2010 | NDBC 44097 | NDBC 44014 NDBC 44056 NDBC 44100 | X | X |
| August 27, 2011 | NDBC 44097 | NDBC 44014 NDBC 44056 NDBC 44100 | X | X |
| October 29, 2012 | NDBC 44097 | NDBC 44014 NDBC 44056 NDBC 44100 | X | X |
| July 4, 2014 | NDBC 44097 | NDBC 44014 NDBC 44056 NDBC 44100 | | X |
| October 9, 2016 | NDBC 44097 | NDBC 44014 NDBC 44056 NDBC 44100 | X | X |
| September 19, 2017 | NDBC 44097 | NDBC 44014 NDBC 44056 NDBC 44100 | X | X |
| September 13, 2018 | NDBC 44097 | NDBC 44014 NDBC 44056 NDBC 44100 NDBC44086 | X | X |
| October 12, 2018 | NDBC 44097 | NDBC 44014 NDBC 44056 NDBC 44100 NDBC44086 | | X |
| September 6, 2019 | NDBC 44097 | NDBC 44014 NDBC 44056 NDBC 44100 NDBC44086 | X | X |

**Table C2.** Available observations of hurricane events in the modeled time period



*Author contributions.* Sarah McElman contributed data, designed the study methodology, and conducted numerical/statistical analysis. Amrit
Verma and Andrew Goupee contributed paper review, editing, and guidance.

*Competing interests.* The authors declare that they have no conflict of interest.

*Acknowledgements.* This work was possible with the valuable insight and perspectives of Louis Bowers, Cyril Frelin, Chan Kwon Jeong, and
Gregory Gerbi. The author would also like to thank the Oceanweather team (Andrew Cox, Erin Harris) for their feedback and for supporting
the use of the GROW-Fine model for academic work.





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
