# Peer review of "Quantifying Tropical Cyclone-Generated Waves in Extreme-Value-Derived Design for Offshore Wind"

_Wind Energy Science, 2024_

## Referee Comment (RC2)

**Review Comment**

December 6th, 2024

Title: Quantifying Tropical Cyclone-Generated Waves in Extreme-Value-Derived Design for Offshore Wind

Author(s): McElman Sarah, Verma Amrit Shankar, and Goupee Andrew

MS No.: wes-2024-129

MS type: Research article

Iteration: Initial submission

**General comment**

This manuscript is discussed about the estimation method of extreme wave height for mixed climate (i.e. the region that both tropical cyclone and extra-tropical cyclone occurs) using different models and concluded with recommended modeling, extreme value analysis method and the number of data years used from extreme analysis for each extra-tropical and tropical cyclone. The conclusion is generally understandable and may help to backup industry's knowledge. However, it is difficult to judge the reasonability of methodologies and conclusions mentioned by authors because there is a lot of missing information.

In conclusion, a reviewer consider that MAJOR REVISION is needed for this manuscript.

**Specific comment**

| Clause/ Subclause | Line number | Comments |
|---|---|---|
| 2 | 79-80 81-85 Figure A | Proper extreme distribution may depend on site characteristic. Although authors discussed about only differences of extreme wave height obtained from Gumbel and Weibull distribution, these result itself doesn't explain the reason that Gumbel distribution is chosen. In other words, why didn't choose Weibull. It is better to draw raw data used for fitting in Figure A. |
| | 85-88 | The authors say "block maxima method was considered suitable for this study" by referring two papers, however, the reason is not clearly mentioned. What part of these papers are referred? Need explanation. |
| | Figure 2 | The word "Calibration", which also appear in text many time |

| | | |
|---|---|---|
| | | is ambiguous. The authors have to explain the detail methodology or procedure. |
| | 96 | Appendix B -> Appendix C |
| | Table 2 | These wave models need not only lateral boundary conditions but also bathymetry or sea surface boundary conditions. Table 1 may be used for sea surface boundary conditions for wave models in Table 2, however, it is difficult to understand it because no explanation made here. Computational area (i.e. domain) for each model is also important information. |
| | | The authors have to explain about these modeling configurations. |
| 2.1.1 | 108 | Is Wrenger (2022) publicly available report? If not, the authors have to explain the relevant part in the report in Annex or somewhere. |
| 2.1.2 | 116 | "vertically nested domain…" is correct? Horizontal nesting to perform locally high-resolution simulation is more common way to use WRF. Explanation about computational domain is needed. |
| | 116 | "real lateral boundary condition…" mentioned here may be CFSR according to Table 1. However, it is difficult to understand that. It is recommended to mention as text clearly. |
| | 120 | Is Georgas(2023) publicly available report? If not, the authors have to explain the relevant part in the report in Annex or somewhere. |
| | 109, 123 | Because 62m and 40m are shallow water region, simulated wave height by wave model is very sensitive to water depth, especially in high wave height. Also, there are geographical distances between buoy and model grid. The authors have to explain the differences between real and modelled water depth. |
| | 129 | Explanation of abbreviation OWI3G is needed. |
| 2.1.3 | 131 | "100 years of tropical storms and 75 years of extra-tropical storms…" Use of as long as data has aspect to improve extreme value, however, old data may have quality |

| | | |
|---|---|---|
| | | problem. The authors have to discuss about data quality issue. |
| 2.1.4 | 147 or Appendix C | There is no explanation about temporal resolution about buoy observation. Also, the authors have to explain how handled or corrected differences of temporal resolutions between each model and buoy observation. |
| | 150, 158 | Unit is needed for the RMSE values. |
| | 143, 151 | The authors have to explain the real and modelled water depth. According to line 109 and 123, water depths are 62m and 40m. Because these depths are shallow, simulated wave heights by wave models are very sensitive to water depth, especially for high wave heights.
Also, according to Table 2 model resolutions are 400m for NA and 600m for MA, however, grid point 29km away from buoy are used for validation. The authors have to explain the reason. |
| 2.2 | 160, 161 | font of "x" in formula and text are different. |
| 2.2.1 | 168, 171 | "exp" and "ln" should not be italic letters. |
| | 169 | "empirical estimation" is not clear explanation. The authors have to explain more detail methodology. |
| | 172 | The authors explain "annual largest value" is used distribution fit. Does tropical cyclone occur and approach to site of interest every year? If not, authors have to explain how handled annual maximum value derived by tropical cyclone for zero tropical cyclone years. |
| | 179 | "Extreme Value Theory assumes that extremes are independent variables." I could understand that what the authors want to say but this sentence may be difficult to understand for some reader. it is suggested to explain a bit detail by changing "extremes are" to other word. |
| | 199-201, Figure 3 | What we can understand from figure 3 is only that extreme distribution obtained from GF models and high-resolutions model show qualitatively close values or distributions for extra-tropical cyclone. Because both these are obtained model, nothing explains storm physics are represented or not. |

| | Figure 3 | Drawing annual maxima used for fitting of extreme distribution in Figure 3 is suggested. |
|---|---|---|
| 3.1.1 | 205 | Dolan-Davis scale and Saffir-Simpson scale are probably US specific. References to explain about these scales are needed. |
| | 210 | Duplicated "the". |
| | 212, 213, Table 1 | Although the authors explain "neither model...", lack of representation of high frequency wave could be caused by frequency range of wave model. Add information about the highest frequency in Table 1. |
| | Figure 4a, c | Why are wave height axis normalized? In general, higher wave height more difficult to simulate. For this reasons, magnitude of wave height is very important information and recommended not to be normalized. |
| | Figure 4d, Figure C1d | Need explanation why SWAN + WRF shows poor Tp resolution. |
| 3.1.2 | 235, 240 | The authors have to explain how inside/outside of storm fetch was defined in this manuscript. Also, explanations about closest approach distance and radius of maximum wind speed of cyclones are need as general information to judge inside/outside. |
| | Figure 5 | Y axis of Figure 5a and 5b are cut off. Also, Tp=18 on Y axis in Figure 5c is missing. |
| | 240-244 | In general, wave periods in inside of storm are dominated by wind-wave and those for outside are significantly affected by wind field both inside and outside of cyclone. Is simulation period enough long, or simulation area enough large? It is suggested that draw wind field and wave height field and add explanation about simulation period about this cyclone. These may help to understand this phenomenon. |
| 3.1.3 | 253-254 | Add reference height of "storm winds (90 knots, or 46 m/s)" |
| | 255 | "Wave buoy measurements occurred on a 30-minute cycle, ..." Is buoy observation available for Hurricane Bob. If so, the authors have to show comparison with observation and modeled value such as Figure 4 and Figure 6b. |

| 3.1.4 | 277-278 | Although the authors considered "primarily to be a function of fetch or duration representation", development of cyclone or error of track depends on model horizontal resolution of weather simulation, or wind data etc. The authors have to understand and explain only wave models are "high resolution" in this study, not for wind models, which were used for input of wave model. |
|---|---|---|
|  | 279-286 | There is no explanation about what Cd model used in this study. Although the authors show wind stress values in Figure 10, these values are strongly affected by Cd models or formula, and each wave models may use different Cd model. Need explanation. |
| 3.2 | 311 | It is not clear that the meaning of "higher-than-average tropical cyclone activity". If it means that annual occurrence is higher than usual, I comment that it is not affect extreme wave height because the authors use only annual maxima. |
|  | 314-315 | The meaning of sentence "in fact..." is not clear. Are extreme distributions in Figure 11a and 12a based on "original high-resolution data set" or "post-processed high-resolution data set"? If the result mentioned in the sentence is not shown in graph, it is better to add word "not shown in graph" in the text. |
|  | Figure 11, Figure 12 | It is suggested to draw annual maxima use for fitting of each extreme distribution to understand reasonability, trend etc. of each distribution. |
| 4 | 342 | "ERA5-boundary conditions" -> "ERA5 wave boundary conditions" is suggested. |
|  | 359-367 | It is understood that these four are main conclusions of this study, however, 1) it is questionable that how "200km or less" in bullet 1 is quantified (e.g. cyclone has radius more then 200km exist.)? There is no detail discussion or deep insight about this. 2) bullet 2-4 are already explained in IEC 61400-1 Annex J. Although the standard is about only for wind, not mentioned for wave, the authors should at least explain and |

| | | |
|---|---|---|
| | | refer in somewhere in this manuscript, then have to explain the differences or originality of this paper. |
| Appendix A | 373 | What value are used for threshold "u"? Need explanation. |
| Appendix C | Figure C1 | Unit for RMSE is needed. Also, definition of NRMSE is needed. |
| | Overall | Because the authors decided to use annual maxima for fitting of extreme value distribution, the validation should be made for annual maximum wave heights. Otherwise, readers don't understand the reasonability of all results and conclusions. |

.

---

## Author Comment (AC3)

The article looks into differences in estimates of significant wave height extremes due to tropical and extra-tropical storms at two offshore locations in the eastern coast of the United States.

Thank you for your comments on this paper, "Quantifying Tropical Cyclone-Generated Waves in Extreme Value-Derived Design for Offshore Wind". Please find below our replies in blue.

The article is poorly written and some of the analyses are not sound. For example:

- There is no motivation being given for the type I tail assumption being made when fitting the Gumbel instead of the Generalized extreme value distribution to the annual maxima. Furthermore, and as can be read in a wealth of extreme value theory publications, for instance the cited book of Coles, the Weibull and the Gumbel distributions are not the asymptotic distributions of data sampled using the peaks-over-threshold approach. This does not mean that they cannot be used but their use should be justified.

  o Thank you. The Peaks-Over-Threshold method is not the selected methodology for analysis in this paper; however, analysis by POT is presented in the appendices for a data subset to investigate any substantial deviations presented by the chosen method itself. This was first investigated by controlling for the same distribution (Gumbel) to assess the influence of the method. In the revised manuscript, the Generalized Pareto distribution is used. No systemic biases are attributed to the selected method over both locations and all datasets investigated (lines 396 – 404).

  o Thank you, we have added justification to the revised manuscript. A one-sample Kolmogorov-Smirnov test was conducted to evaluate whether each of the four data sets (North Atlantic high-resolution, Mid-Atlantic high-resolution, GF-EC Tropical in the N. Atlantic and GF-EC Tropical in the Mid-Atlantic) follow a Gumbel cumulative distribution function. The null hypothesis ($H_o$: the data follows a Gumbel distribution) was not rejected, indicating that, at a 95% confidence level, the Gumbel distribution fits the data adequately (see figures on the following page):

Internal Use

[Figure]

[Figure]

[Figure]

Internal Use

[Figure]

- Also, the authors confuse the directional spreading of a wave system or sea state with the variability of the mean wave direction during a storm.

    o Thank you, the details may not have been apparent in the original manuscript, but we will clarify: Directional spreading describes the radial propagation of wave energy, in this context due to applied forcing (wind). Here, the observed features of radial wave propagation—spread—based on the energy flux from winds to ocean surface (generated by a tropical or extra-tropical cyclone) relies heavily on the nature of the flux (storm type), consistent with the physical system described by Forristall and Ewans ("Worldwide Measurements of Directional Wave Spreading", Journal of Atmospheric and Oceanic Technology, 1998). Within the storm fetch, it is not a given that there are many sea states, but instead are highly dominated by local winds. As addressed in the calibration section for the Mid-Atlantic Bight model, wave boundaries with applied ERA-5 wave data on all 3 sides performed worse for wave direction than when forced on only one boundary by ERA-5 and with open boundaries on the remaining two sides. This is attributed to the better performance of modeled wind-driven waves than prescribed directly by the global reanalysis dataset. This investigation was therefore considered necessary when gauging the physical representation by the models.

Specific comments

Lines 82-83: What is the rational for fitting the Weibull (of minima, I assume) and the Gumbel distribution to POT data? Can you justify why you are deviating from the Generalized Pareto distribution?

Internal Use

- The peaks-over-threshold method is not focus of paper, but as described above, the inclusion of this detail is to gauge result sensitivity to the paper method, rather than to determine the single best fit for each data set. As GP is traditionally the best selection for POT analysis, Figures A2a and A2b have been updated with return value estimates from GP distribution over the threshold indicated in the legend.

Lines 136-138: Please indicate whether there is corresponding between the storms leading to the annual maxima in both GF and hindcast datasets.

- Thank you, the question is not entirely clear. If the reviewer means "correspondence", the investigated peaks are associated with historical tracks for each storm. For each storm, lines 179 – 182 of the original manuscript (lines 185 – 186 of the revised manuscript) state: *"To preserve the independence criterion in this study, only the peak significant wave height is retained in a period of 98 hours during an identified storm."*

Lines 159-161: Please rephrase of remove. The estimates are obtained using the likelihood method? If so, it suffices to state it.

- The values are determined by maximum likelihood; the text is updated (line 168 of the revised manuscript) for clarity.

Section 2.2.1: Please motivate why the Gumbel instead of the Generalized extreme value distribution is being fitted to the data.

- A one-sample Kolmogorov-Smirnov (KS) test was conducted to evaluate whether the data follows a Gumbel cumulative distribution function. The null hypothesis ($H_o$: the data follows a Gumbel distribution) was not rejected at the 5% significance level, indicating with 95% confidence that the Gumbel distribution fits the data adequately. (Please refer to the plots on pages 2 and 3 of this reply.)

  Additionally, calculated GEVD parameters for the four main datasets in this work have a shape factor, k, that is close to or effectively 0:

| | location - mu | scale - sigma | shape - k |
|---|---|---|---|
| NA GF-EC: *GEVD* | 4.3879 | 2.122 | -0.0498 |
| NA GF-EC: *Gumbel* | 4.31 | 2.17 | 0 |
| | | | |
| NA HiRes: *GEVD* | 6.564 | 0.92 | 0.192 |
| NA HiRes: *Gumbel* | 6.57 | 1.34 | 0 |
| | | | |
| MAB GF-EC: *GEVD* | 4.624 | 2.53 | -0.2168 |
| MAB GF-EC: *Gumbel* | 4.42 | 1.93 | 0 |

Internal Use

| | | | |
|---|---|---|---|
| MAB HiRes: *GEVD* | 5.7588 | 0.7988 | 0.1351 |
| MAB HiREs: *Gumbel* | 5.49 | 1.34 | 0 |

This data is also provided in Appendix B, revised Table B1.

Both of these factors support the use of a Gumbel distribution for this study.

Lines 170-171: Why is *p* called "probability **period**"? Please add that when computing the return values p is substituted by 1/*n* with *n* being the return period in years. In the tables and text only *n* is being given, not *p*.

- The term has been updated to "Annual exceedance probability", which is p = 1/n

Appendix B. The associated return value, $x_p$, for annual exceedence probability $p$ are calculated as:

$$\quad x_p = \mu + \sigma\left(-\ln(-\ln(1-p))\right), \quad for \quad -\infty < x_p < \infty$$

Line 178: Why does the storm list given in Appendix C only starts in 1991?

- The storm list is provided for the hindcast period with overlapping observations. The MAB hindcast starts in 1990 and the first observations in region are in 1991. Observations do not occur in NA until 2009. Lines 93 – 94 of the revised manuscript are updated to: *"A list of significant storms during the hindcast period, available observations, and storm events used for model calibration is provided in Appendix C, beginning with the first available buoy-based observations."*

Figure 3: 1)The data to which the distributions were fitted need to be added to the figure. (If not possible in absolute scale, then in relative scale as in Figure 4.) 2) Preferably also the 95% confidence intervals of the estimates should also be given. 3)The legend should contain for each of the lines the periods covered by the data or the sample size (number of considered annual maxima).

- Thank you. The fits are for the entire period outlined in Tables 1 and 2 unless otherwise indicated in the legend (i.e., only for Figures 13 and 14 in the revised manuscript). Lines 179 – 185 of the revised manuscript state: *"The GF-EC MAB site data includes 81 tropical cyclone events. The GF-EC NA site data includes 80 tropical cyclone events. In the event that a tropical cyclone did not pass nearby the analysis location during the season, no maxima are recorded for that year in the GF-EC dataset. In the "tropical cyclone only, high-resolution" NA and MAB datasets, a smaller, non-extra-tropical cyclone is picked as a maxima for a year without nearby tropical cyclones activity. Annual maxima analysis for the "high-resolution" model datasets account for 42 storms (tropical or extra-tropical*

*cyclones) at the NA location and 30 storms (tropical or extra-tropical cyclones) at the MAB location."*

Figure 4 of the revised manuscript now presents the data and Gumbel distribution for the four main datasets investigated:

[Figure]

*Figure 1: MAB GF-EC Tropical*

[Figure]

*Figure 2: NA GF-EC Tropical*

[Figure]

*Figure 3: NA high-resolution model*

[Figure]

*Figure 4: MAB high-resolution model*

A one-sample Kolmogorov-Smirnov (KS) test was conducted to evaluate whether the data follows a Gumbel cumulative distribution function. The null hypothesis ($H_0$: the data follows a Gumbel distribution) was not rejected at the 5% significance level, indicating with 95% confidence that the Gumbel distribution fits the data adequately. (Please refer to the plots on pages 2 and 3 of this reply.)

Internal Use

Line 91: There are only 15 samples in the 'GF-EC. Trop.' fit? Please comment on the uncertainty of the estimates.

- We agree that 15 events are a small sample size for calculating extreme values. There are 15 identified hurricanes during the "NA high-resolution" hindcast period (1979 – 2021). There are 80 in the NA GF-EC Tropical period (1924 – 2021).

Section 3.1.2: In my opinion this section can be removed. What is its purpose? Why are the plots of the significant wave height (even if normalised) not shown?

- The performance of the numerical models in this section are investigated: *Is poor numerical representation of the sea state a cause of differences in return values?* Given that some global reanalysis datasets have been shown to under-represent the highest winds from tropical cyclones, Section 3.1.2 shows that, through overall representation Hs, the partitioned sea states, and wave trajectory over the course of the storm, there is fairly good consistency despite potentially high variability for tropical cyclones as compared to extra-tropical cyclones. Numerical performance is an essential piece in assessing how extremes are represented, and to what degree calibration can improve validation (Neary (2020), Caires (2005), Stephens (2006)), with the goal of estimating more suitable return values. It may be surprising to some readers that generous calibration of values was not enough to close the gap between the trend and magnitude of estimated return values between the data sets, even over the same time period (storm sample size). This has nontrivial implications for the design of metocean models in hurricane-prone regions, as direct modeling of TCs or synthetic representation of TCs are not required by standards for ocean models. Out of consideration for the total number of figures in the paper, a selection of events with measurements were originally chosen. Bob (no observations) and Dorian for the North Atlantic have been added to the revised manuscript Figure 6.

Section 3.1.3: The contents of this section are incorrect. First, how can the authors not be aware that the waves in the roses in Figure 7 are from the coast and therefore not realistic. Second, the authors present the variation in the mean wave directions during the consider storms (may wave systems, sea states) and analyse with reference to the article of Forristall and Ewans on directional spreading of a wave system or sea state.

- Thank you for pointing this out. During the submission process, we identified this error and posted a WESD comment to this effect when the manuscript became public (Nov 4[th]). A scripting error in the rose plot code shifted measurements clockwise by 90 degrees, and the images have been revised. Note that the analysis location is ~90 km offshore from the west. Your 2[nd] comment is answered on page 1 of this reply sheet.

Section 3.2: This section needs also to be completely redone. When comparing statistical estimates the sample sizes and confidence intervals should be given.

Internal Use

Furthermore, when making assumption in terms of the tail of the data these should be justified.

- Thank you, these points are addressed in replies to your previous comments. Figures of the data and fit, are added to the manuscript (revised Figure 4), with 95% confidence intervals in Table B1.

Technical corrections

Line 46:  Please specify which are the variables being considered in the univariate and bivariate analyses you are referring to. Why is this relevant for this article?

- The paper focuses on univariate extreme significant wave height, which is an approach commonly found in the literature. However, it can be more realistic to consider seas in terms of environmental contours—bi-variate analysis—especially when it comes to a small, focused storm event like a hurricane. While investigation of the actual differences in return values between uni- and bi-variate return value assessments during different types of events is worthwhile, it is beyond the scope of this paper. This is mentioned in the paper only to clarify and ensure that uni-variate analysis does not misrepresent significant wave height as it is projected to larger return periods. Specification that the return values in this paper are determined with a univariate approach is specified in the revised manuscript (Section 2.2).

Line 69: Specify which 3 models and models of what?

- Line 69 in the original manuscript (line 70 of the revised manuscript) refers to the 3 numerical ocean models, called out for both regions in Figure 1, and described in Tables 1 (wind) and 2 (waves + hydrodynamics) in the column "Model".

Line 78: What does "Return period results" mean? Should it be "Return value estimates"?

- "Results" has been changed to "estimates".

Line 96: You mean Appendix C instead of B?

- Thank you, this error has been updated.

Line 104: Is the magnitude of  Cds correct? Please introduce the meaning of the symbols it wanting to give the values.

- Thank you, the coefficient of surface drag, C_ds, was missing a factor of 10^(-2) and is updated. The names of these wave model parameterizations are added.

Internal Use

Line 110 and elsewhere: Explain what you mean with "most "at-risk" turbine location" and how this has been defined.

- This is identified for the present study as the position most exposed to waves from Atlantic Basin combined with deepest points in the Wind Energy Area. Given that this is not a significant detail for this investigation, it has been updated to "turbine location of interest" in the revised manuscript (lines 106, 118).

Lines 114-118, ...: Provide references for SWAN, Delft3D, Westhuysen, WAM, OWI3G,...

- These citations have been added to the revised manuscript.

Line 117: Define acronyms throughout the text. For instance, what does YSU mean?

- YSU is the "Yonsei University" scheme, a non-local turbulence closure planetary boundary layer model in WRF. This specification has been added to the revised manuscript in line 113.

Lines 124: State also model depth for the considered output location.

- The depths of the points of interest/analysis are given in the original manuscript model descriptions, lines 109 and 123 (revised manuscript lines 105, 117). It is specified in the revised manuscript that the nearby GF-EC analysis points are located at the same depths (lines 107, 119).

---

## Author Comment (AC4)

**General Comment**

This article proposes a methodology to find extreme wave values in regions where both tropical and extra-tropical cyclones occur. The motivation given is for the more accurate prediction of design values for offshore wind turbine farms. Long term return values (up to 10,000 year return periods) are calculated using statistical methods. Overall, I find that this article has scientific merit; however, the findings/methods are somewhat obfuscated/unclear. I believe that by clarifying the methodologies used and providing more context for why specific scientific decisions were made the article would be significantly improved.

Thank you for the time and effort you dedicated to reviewing this paper, "Quantifying Tropical Cyclone-Generated Waves in Extreme Value-Derived Design for Offshore Wind". Please find below our replies in blue to your comments.

**Specific Comments**

Line 73: You mention "validated and calibrated" models. Have these calibrations/validations been published elsewhere? If so, please include the citation to the appropriate papers. If not, please include a subsection with an overview of both the calibration (what was calibrated, how parameters were selected, etc.) and some of the validation data.

- Thank you. The referenced model reports are not public (and the references have been removed), however, the relevant details are re-created and expanded in the revised manuscript "Model Skill" section (*"The North Atlantic model calibration was determined from a range of cap to friction velocity values and nonlinear growth coefficients for overall performance during 1) a mixed set of storms, and 2) during the entire year of 2012"*)

  and in Appendix C (*"The Mid-Atlantic model was calibrated against LiDAR buoy observations of "Hurricane Isais" in July 2020. The friction coefficient, whitecapping parameterization (Westhuysen or Komen), time step, and wave boundary conditions were adjusted during the calibration process. The boundary condition calibration (ERA-5 wave forcing on three boundaries, or ERA-5 waves on the eastern boundary with open condition on the remaining boundaries) resulted in notable differences in the mean wave direction compared to observations during hurricane passage. This is attributed to improved treatment of wind-driven waves during this event, and ERA-5 forcing only on the eastern boundary was chosen for the final configuration. Significant wave heights were similar to observations for both boundary condition configurations."*):

| Location and Duration | Hs [m] | | | T02 [s] | | | Tp [s] | | |
|---|---|---|---|---|---|---|---|---|---|
| | Correlation Coefficient [%] | Scatter Index | Bias [m] | Correlation Coefficient [%] | SI | Bias [m] | Correlation Coefficient [%] | SI | Bias [m] |
| NDBC44097 40.97 °N, 71.12 °W 2009–2020 | 91 | 0.25 | 0.03 | 82 | 0.13 | -0.52 | 73 | 0.23 | -0.01 |
| NDBC44008 40.50 °N, 69.25 °W 1982–2020 | 93 | 0.24 | -0.03 | 82 | 0.12 | -0.39 | 75 | 0.17 | 0.06 |
| NDBC44017 40.69 °N, 72.05 °W 2002–2020 | 92 | 0.25 | 0.10 | 83 | 0.11 | -0.38 | 77 | 0.20 | 0.13 |
| NDBC44020 41.50 °N, 70.28 °W 2009–2020 | 81 | 0.35 | 0.10 | 61 | 0.07 | -0.65 | 44 | 0.11 | -0.15 |
| LiDAR Buoy 41.07 °N, 70.48 °W 2018–2020 | 91 | 0.26 | 0.08 | 84 | 0.11 | -0.21 | 71 | 0.22 | 0.09 |

**Table 3.** Validation statistics throughout the North Atlantic model domain against continuous measurements, 2010 - 2020.

| Location | Hs [m] | | | Tp [s] | | |
|---|---|---|---|---|---|---|
| | Correlation Coefficient [%] | Scatter Index | Bias [m] | Correlation Coefficient [%] | SI | Bias [m] |
| Combined 20 Buoys | 0.93 | 0.24 | 0.13 | 0.59 | 0.28 | -0.07 |
| NDBC44008 40.50 °N, 69.25 °W | 0.90 | 0.21 | 0.04 | 0.47 | 0.32 | -0.95 |
| NDBC41025 35.01 °N, 75.45 °W | 0.90 | 0.23 | 0.20 | 0.59 | 0.24 | -0.27 |

**Table 4.** Selected validation statistics for the GROW-Fine East Coast model, overall and in the North and Mid-Atlantic regions.

| | Whitecapping | Time step | Chezy coefficient | Jonswap coefficient | |
|---|---|---|---|---|---|
| Calibration Range | Westhuysen or Komen | 2min–0.2 min | 65–85 | 0.25–0.85 | |
| Selected | Westhuysen | Variable, reduced from 2 min during certain events | 65 | 0.67 | |

**Table C1.** Calibration values for the MAB "high-resolution" model.

| Location | RMSE | Mean Absolute Error |
|---|---|---|
| LiDAR Buoy 06/2020 - 07/2020 | 0.26m | 0.20m |
| NDBC44099 2009–2019 | 0.25m | 0.18m |

**Table C2.** Validation statistics for the MAB "high-resolution" model.

Line 80: Why was the Block Maxima with Gumbel Fit selected over, for example, a Generalized Pareto DIstribution with peaks over threshold? Just below this line you mention that you performed a sensitivity analysis using different methods but you do not explain why you ended up highlighting the BM with Gumbel results.

- Thank you, this is important to clarify. The two main methods for evaluating extremes for offshore applications are considered to be POT and BM (ref. Jonathan 2013). As elaborated in the following reply, POT was not considered suitable for accurately

representing the GF-EC data set. However, the POT sensitivity study was updated to include return estimates by Generalized Pareto in the revised manuscript:

[Figure]

[Figure]

(a) Return values at the North Atlantic site from the high-resolution model ("Combined") and two post-processed subsets of the high-resolution model based on storm type.

(b) Return values at the Mid-Atlantic Bight site from the high-resolution model ("Combined") and two post-processed subsets of the high-resolution model based on storm type.

**Figure A2.** Return values by Peaks-Over-Threshold for the "high-resolution" return estimates with a Generalized Pareto distribution and selected thresholds, $u$.

A one-sample Kolmogorov-Smirnov test was conducted to evaluate whether the data follows a Gumbel cumulative distribution function. The null hypothesis ($H_0$: the data follows a Gumbel distribution) was not rejected, indicating that, at a 95% confidence level, the Gumbel distribution fits the data adequately. While the Generalized Extreme Value Distribution is the most comprehensive distribution choice for BM analysis, the GEVD shape parameter is near or effectively 0 for the four datasets considered in this paper and priority was given to the treatment of the distribution tail. The GEVD parameters are added to the manuscript appendix:

| | location - mu | scale - sigma | shape - k |
|---|---|---|---|
| NA GF-EC: *GEVD* | 4.3879 | 2.122 | -0.0498 |
| NA GF-EC: *Gumbel* | 4.31 | 2.17 | 0 |
| | | | |
| NA HiRes: *GEVD* | 6.564 | 0.92 | 0.192 |
| NA HiRes: *Gumbel* | 6.57 | 1.34 | 0 |
| | | | |
| MAB GF-EC: *GEVD* | 4.624 | 2.53 | -0.2168 |
| MAB GF-EC: *Gumbel* | 4.42 | 1.93 | 0 |
| | | | |
| MAB HiRes: *GEVD* | 5.7588 | 0.7988 | 0.1351 |
| MAB HiREs: *Gumbel* | 5.49 | 1.34 | 0 |

Line 89:  Could you explain more about why a POT method is not appropriate for "only storm events"? Given that POT assumes the events are independent (which I would say applies to individual storm events) and that threshold selection, whether through graphical or automated methods, relies upon the fact that for any threshold that produces an adequate fit a threshold larger than that should produce the same fit (when using a generalized Pareto distribution), I fail to see why the lack of "normal sea states" precludes the use of a POT methodology.

- Yes, thank you—determining an "adequate fit" is key. The GF-EC data sets are non-continuous time series of storms over roughly 8-day periods. Despite covering 100 years of events, in the case of the tropical cyclone model, the relatively short individual storm period meant threshold and clustering time selection were based on this limited range (the growth and decay of a storm peak) challenging whether the assessed fits were "adequate". (This detail is added to the revised manuscript in line 169.) While trying to maintain a sufficient data sample for fitting, the calculated return values by this method for the GF-EC model data were suspiciously large, suggesting that the fitted distribution did not fully characterize the site.

Section 2.1: I find the description of the numerical models to be lacking in detail. As you mention in multiple locations, the location and derivation of boundary conditions can greatly change the results of a numerical model. Despite this, there is no description of the model domains, i.e., does the model cover the entire North Atlantic basin? Does it only cover the insets from Figure 1? You also mention again that the models are "locally validated". Where can I find this validation data? You mention the Commonwealth Wind metocean report (Wrenger, 2022) at line 108. Using the information in your works cited, I was unable to locate this report. There is the same issue with the Georgas (2023) report you cite for the Mid-Atlantic model (line 122). For the GROW-Fine East Coast model we are simply referred to Oceanweather inc. Please either provide the validation statistics in your work or, if possible, provide open-source and easily accessible reports showing why we should trust these models.

- Thank you, these are important details for assessing model applicability and validity for this study. Validation statistics have been added to the manuscript (Tables 3, 4, and C2 shown above). The model domains and validation locations have been added in Figure 2

to the revised manuscript:

[Figure]

**Figure 3.** Analysis locations are indicated by circles and validation locations are indicated by triangles. Clockwise from the left: (a) The structured-grid GF-EC domain spans from 25 to 45.85 °N, and 82 to 64.3 °W. (b) The unstructured-grid NA "high-resolution" domain spans from 39 to 41.5 °N, and 73 to 68 °W. The wave boundary conditions are taken from a regional spectral wave model that spans 28 to 46 °N, and 82 to 58 °W, covering 16 directions and 25 frequencies from 1 to 33s. (c) The structured-grid MAB "high-resolution" domain spans from 35.83 to 37 °N, and 75.58 to 74.83 °W.

As the model reports are not publicly available, and these models were designed and developed for internal industrial use, they are removed from the references.

Section 2.1.4: I see here some mention of model validation. Consider moving some part of Appendix C into the body of the text. Especially given you specifically refer to the figures and error values in the appendix it seems appropriate that it would be part of the main text.

- Yes, there was some back-and-forth about the best location for this. The quantile-quantile plots that previously appeared in Appendix A have been replaced by NA and GF-EC model validation statistics in the "Model Skill" section (revised Tables 3 and 4) and MAB validation statistics in Appendix C (Table C2).

Figures 11 and 12: What are the confidence intervals of the return periods you calculate here? Given the use of m such a short time series for the estimation of very long return periods I would expect to see relatively large confidence intervals.

- Thank you. For clarity, Figures 11 and 12 are maintained as presented in the original manuscript. 95% confidence intervals are added to the appendix in Table B1.

There are other locations where the methodology could be clarified and greatly improve this manuscript. At the moment, I find that the experiments herein would be very difficult for another researcher to reproduce, greatly limiting the usefulness of the findings.

- Thank you, the manuscript has been updated to better reflect these details in your above comments, and if any important details remain missing, we would be happy to further revise.

  Regarding your question on the reproducibility of our results and methods, this paper attempts to elucidate industry-standard methods and tools, which are often kept proprietary, for public discussion and scrutiny. Furthermore, the "high-resolution" models presented here are examples we've selected to represent typical tools and robust methods in common practice today. We believe this is important for improving the state of the art in the industrial, standards-development, and academic domains (where numerous recent publications have calculated Hs extremes for this region from a single "mixed-type" sample, for example). The full wave timeseries for the NA and MAB locations, in addition to nearby observational data, are publicly available at https://doi.org/10.5281/zenodo.13884957. The GROW-Fine East Coast model is the only long-duration, direct-hindcast tropical and extra-tropical model we are aware of that allows direct comparison; while we are unable to publish long timeseries or peak absolute values from this dataset, we believe that the clear trends discussed in this work are useful for current and future model development and infrastructure design activities.

---

## Author Comment (AC5)

Thank you for the time and effort you dedicated to reviewing this paper, "Quantifying Tropical Cyclone-Generated Waves in Extreme Value-Derived Design for Offshore Wind". Your comments have helped to improve the quality of this manuscript. Please find below our replies in blue.

1. Proper extreme distribution may depend on site characteristic. Although authors discussed about only differences of extreme wave height obtained from Gumbel and Weibull distribution, these result itself doesn't explain the reason that Gumbel distribution is chosen. In other words, why didn't choose Weibull. It is better to draw raw data used for fitting in Figure A.

    Thank you. The annual maxima data with a Gumbel distribution are added to the manuscript (Figure 4 from the revised manuscript, reproduced below). Confidence intervals are added to the Appendix in Table B1. A one-sample Kolmogorov-Smirnov test was conducted to evaluate whether the data follows a Gumbel cumulative distribution function. The null hypothesis ($H_0$: the data follows a Gumbel distribution) was not rejected, indicating that, at a 95% confidence level, the Gumbel distribution fits the data adequately. Please also see the response to Question 1 of the Reviewer 1 comments.

[Figure]

*Figure 1: MAB GF-EC Tropical*

[Figure]

*Figure 2: NA GF-EC Tropical*

[Figure]

*Figure 3: NA high-resolution model*

[Figure]

*Figure 4: MAB high-resolution model*

2. The authors say "block maxima method was considered suitable for this study" by referring two papers, however, the reason is not clearly mentioned. What part of these papers are referred? Need explanation.

   a. Papers by Bhaskaran (*Comparison of Extreme Wind and Waves Using Different Statistical Methods in 40 Offshore Wind Energy Lease Areas Worldwide*, Energies, 2023), Barthelmie (*Extreme Wind and Waves in U.S. East Coast Offshore Wind Energy Lease Areas,* Energies, 2021), and Jonathan (*Statistical modelling of extreme ocean environments for marine design: A review*, Ocean Engineering, 2013) are referenced as published examples of suitable applications of Annual Maxima for significant wave height extreme value analysis. Additional figures for the Block Maxima-Gumbel distribution are provided in the revised Figure 4, and in the appendix (KS one-sample tests).

[Figure]

[Figure]

[Figure]

[Figure]

3. The word "Calibration", which also appear in text many time is ambiguous. The authors have to explain the detail methodology or procedure.
   a. The calibration and "model skill' (validation) discussions have been revised with statistics and more details on the validation locations, time periods, storms used, and calibration parameters. Calibration of the North Atlantic model is discussed in "Model Skill", line 143: *The North Atlantic model calibration was determined from a range of cap to friction velocity values and nonlinear growth coefficients for overall performance during: 1) a mixed set of storms, and 2) during the entire year of 2012. After this calibration, validation was conducted over a 10-year period at five observation locations throughout the model domain.*

   More details on the Mid-Atlantic calibration are also provided in Appendix C.
4. Appendix B -> Appendix C
   a. Thank you, updated.
5. These wave models need not only lateral boundary conditions but also bathymetry or sea surface boundary conditions. Table 1 may be used for sea surface boundary conditions for wave models in Table 2, however, it is difficult to understand it because no

explanation made here.  Computational area (i.e. domain) for each model is also important information. The authors have to explain about these modeling configurations.

  a. Thank you. The bathymetric input to each model is added to the model descriptions in sections 2.1.1, 2.1.2, and 2.1.3, along with domain extents, which appear in Figure 3 in the revised manuscript:

[Figure]

**Figure 3.** Analysis locations are indicated by circles and validation locations are indicated by triangles. Clockwise from the left: (a) The structured-grid GF-EC domain spans from 25 to 45.85 °N, and 82 to 64.3 °W. (b) The unstructured-grid NA "high-resolution" domain spans from 39 to 41.5 °N, and 73 to 68 °W. The wave boundary conditions are taken from a regional spectral wave model that spans 28 to 46 °N, and 82 to 58 °W, covering 16 directions and 25 frequencies from 1 to 33s. (c) The structured-grid MAB "high-resolution" domain spans from 35.83 to 37 °N, and 75.58 to 74.83 °W.

6. Is Wrenger (2022) publicly available report? If not, the authors have to explain the relevant part in the report in Annex or somewhere.
    a. Thank you. Unfortunately, the project report is not a publicly-available document, however, relevant details concerning modeling, validation, and study reproducibility are incorporated in the revised paper. The reference is removed.
7. "vertically nested domain…" is correct? Horizontal nesting to perform locally high-resolution simulation is more common way to use WRF. Explanation about computational domain is needed.
    a. Thank you, this is a typo and has been updated to "horizontally-nested".
8. "real lateral boundary condition…" mentioned here may be CFSR according to Table 1. However, it is difficult to understand that. It is recommended to mention as text clearly.
    a. "Real lateral" is a WRF boundary condition option, as opposed to an idealized applied value; it is specified that this refers to the CFSR model data (line 113).

9. Is Georgas(2023) publicly available report? If not, the authors have to explain the relevant part in the report in Annex or somewhere.
   a. Thank you. Unfortunately, the project report is not a publicly-available document, however, relevant details concerning modeling, validation, and study reproducibility are incorporated in the revised paper. The reference is removed.
10. Because 62m and 40m are shallow water region, simulated wave height by wave model is very sensitive to water depth, especially in high wave height. Also, there are geographical distances between buoy and model grid. The authors have to explain the differences between real and modelled water depth.
    a. The real depth at buoy 44097 is 49.4m. The modeled depth at the NA high-resolution analysis point is 62m. The real depth at buoy 44014 is 49.1m. The modeled depth at the MAB high-resolution analysis point is 38m. The manuscript has been updated to reflect this (section "Model descriptions"), with bathymetric sources.
11. Explanation of abbreviation OWI3G is needed.
    a. The "Oceanweather 3$^{rd}$ generation" description is added to the text (line 124).
12. "100 years of tropical storms and 75 years of extra-tropical storms..." Use of as long as data has aspect to improve extreme value, however, old data may have quality problem. The authors have to discuss about data quality issue.
    a. Thank you, this is an important point. The model is both validated and verified for post-1979 events. Selected events are then simulated with the verified model based on available observations, however limited, or to events which were known to have a significant impact on coastal areas. The text is updated to reflect this.
13. There is no explanation about temporal resolution about buoy observation. Also, the authors have to explain how handled or corrected differences of temporal resolutions between each model and buoy observation.
    a. Buoy temporal resolution is added to the text. Note that the quantile-quantile plots (formerly Appendix A) have been replaced by validation statistics (Tables 3, 4, and C2 in the revised manuscript) for all three models. Please see replies to Reviewers 1 and 3 for examples of these validation tables.
14. Unit is needed for the RMSE values.
    a. RMSE unit of m is added. The figures have been replaced with tabulated values from multiple buoys in Tables 3, 4, and C2 in the revised manuscript. Please see replies to Reviewers 1 and 3 for examples of these validation tables.
15. The authors have to explain the real and modelled water depth. According to line 109 and 123, water depths are 62m and 40m. Because these depths are shallow, simulated wave heights by wave models are very sensitive to water depth, especially for high wave heights. Also, according to Table 2 model resolutions are 400m for NA and 600m for MA, however, grid point 29km away from buoy are used for validation. The authors have to explain the reason.
    a. All buoys in the region have been used for validation and a selection for calibration, as shown in the new Figure 3 (see comment 5). High-resolution geophysical survey data for each turbine location (that is, both analysis points

discussed here) were an input to the metocean model, and supplemented by GEBCO data outside of the project area, which has been specified in the model descriptions.

16. font of "x" in formula and text are different.
    a. Updated formatting.
17. "exp" and "ln" should not be italic letters.
    a. Updated formatting
18. "empirical estimation" is not clear explanation. The authors have to explain more detail methodology.
    a. The text now clarifies that the estimation procedure is maximum likelihood (line 169)
19. The authors explain "annual largest value" is used distribution fit. Does tropical cyclone occur and approach to site of interest every year? If not, authors have to explain how handled annual maximum value derived by tropical cyclone for zero tropical cyclone years.
    a. In the Mid-Atlantic Bight, tropical cyclones are an annual occurrence. In the North Atlantic, there are a handful of years where tropical cyclones did not occur. In the post-processed continuous models, where extra-tropical events have been removed, a smaller, non-TC, non-ETC event is picked as a maxima (TCs and ETCs are the strongest storms in the region). In the GF-EC dataset, no maxima is selected for the year. The text is updated to reflect this (line 173).
20. "Extreme Value Theory assumes that extremes are independent variables." I could understand that what the authors want to say but this sentence may be difficult to understand for some reader. it is suggested to explain a bit detail by changing "extremes are" to other word.
    a. Updated phrase to "extreme values are"
21. What we can understand from figure 3 is only that extreme distribution obtained from GF models and high-resolutions model show qualitatively close values or distributions for extra-tropical cyclone. Because both these are obtained model, nothing explains storm physics are represented or not.
    a. Yes, Figure 3 only represents the statistical distributions. However, the clear difference in trend and magnitude motivates further investigation into how these storms are represented (i.e., if physical processes are well/poorly resolved) in the following sections.
22. Drawing annual maxima used for fitting of extreme distribution in Figure 3 is suggested.
    a. Thank you. This has been addressed in previous comments and is added to the manuscript.
23. Dolan-Davis scale and Saffir-Simpson scale are probably US specific. References to explain about these scales are needed.
    a. The text has been updated with originating references.
24. Duplicated "the".
    a. Thank you, this is updated.

25. Although the authors explain "neither model…", lack of representation of high frequency wave could be caused by frequency range of wave model. Add information about the highest frequency in Table 1.
    a. Thank you, this was erroneously left out in the original manuscript. The maximum frequency for each model is added to Table 2 of the revised manuscript:

| Model | Tool | Resolution | Boundary Conditions | Coupling | Spectral Parameterization |
|-------|------|-----------|---------------------|----------|---------------------------|
| NA HiRes | MIKE21 | 600m wave (2D) 600m hydro (2D) 1-hour | DHI East Coast Waves (waves) DHI East Coast (hydro) | 1-way, hydro to waves | 36 directions 32 freq. bins 0.033 Hz min 0.667 Hz max |
| MAB HiRes | SWAN+DELFT3D | 400m wave (2D) 400m hydro (3D) 1-hour | ERA5 (waves) HYCOM (hydro) | 2-way, waves and hydro | 36 directions 24 freq. bins 0.05 Hz min 0.448 Hz max |
| GROW-Fine East Coast | OWI3G+ADCIRC | 5.5km wave (2D) 5.5km hydro (2D) 15-minute | GROW2012 (waves) Prevost '08 (hydro) | No dynamic coupling. Reanalysis of each modeled storm. | 48 directions 26 freq. bins 0.029 Hz min 0.322 Hz max |

**Table 2.** Wave and hydrodynamic parameterization for the three investigated models in the North Atlantic, Mid-Atlantic Bight, and along the US Atlantic coast (GROW-Fine East Coast). All model bathymetries are derived from GEBCO. The NA and MAB model bathymetries are supplemented with 1m geophysical survey measurements within the project area.

26. Why are wave height axis normalized? In general, higher wave height more difficult to simulate. For this reasons, magnitude of wave height is very important information and recommended not to be normalized.
    a. We understand that absolute values are important for quantifying the quality of a modeled event. However, as a condition for use of the GF-EC dataset for academic investigation (safeguarding the intellectual property of the GF-EC product) the normalized results are presented here, to preserve scale between model results. We believe that this observable difference in scale is nonetheless valuable, as there are no other comparable datasets publicly available.
27. Need explanation why SWAN + WRF shows poor Tp resolution.
    a. The quantile-quantile plot of Tp (MAB "high-resolution") this comment refers to has been replaced by a set of validation statistics in Tables 3, 4, and C2 of the revised manuscript. Please see the replies to Reviewers 1 and 3 for the revised validation tables.
28. The authors have to explain how inside/outside of storm fetch was defined in this manuscript. Also, explanations about closest approach distance and radius of maximum wind speed of cyclones are need as general information to judge inside/outside.
    a. Inside/outside categorization was taken here simply as within or beyond of 200 km from the storm eye. This was identified for the closest point on the storm IBTRaCS record to the analysis location. Two similarly-scaled events with clear differences in distance were selected for this purpose.
29. Y axis of Figure 5a and 5b are cut off. Also, Tp=18 on Y axis in Figure 5c is missing.
    a. Thank you, the formatting is updated.
30. In general, wave periods in inside of storm are dominated by wind-wave and those for outside are significantly affected by wind field both inside and outside of cyclone. Is simulation period enough long, or simulation area enough large? It is suggested that

draw wind field and wave height field and add explanation about simulation period about this cyclone. These may help to understand this phenomenon.

    a. Thank you. More details about the model domains are added in Figure 3 of the revised manuscript. The high-resolution models are continuous on an hourly basis. The GF-EC model timescale varies between individual events and concludes when the storm peak decreases to pre-storm levels; for example, the GF-EC full storm period for the three events is shown in Figure 7 of the revised manuscript. While the GF-EC model may only cover a snapshot of the storm as it passes by a location, the domain extent allows the full storm extent on the east coast to be modeled, from growth through decay.

31. Add reference height of "storm winds (90 knots, or 46 m/s)"

    a. The value is taken from IBTRaCS; the US National Hurricane Center defines the surface wind speed as 1-minute sustained average at a 10m height. The manuscript is updated to reflect this (line 255).

32. "Wave buoy measurements occurred on a 30-minute cycle, ..." Is buoy observation available for Hurricane Bob. If so, the authors have to show comparison with observation and modeled value such as Figure 4 and Figure 6b.

    a. Unfortunately not. Observations began at (NA location) NDBC 44097 in 2009. Additionally, there is no recorded data at (MAB location) NDBC 44014 during the passage of Bob.

33. Although the authors considered "primarily to be a function of fetch or duration representation", development of cyclone or error of track depends on model horizontal resolution of weather simulation, or wind data etc. The authors have to understand and explain only wave models are "high resolution" in this study, not for wind models, which were used for input of wave model.

    a. Thank you, this is a helpful clarification. The "high-resolution" nomenclature was one way to identify a commonality for investigating wave model performance. A specification that the high-resolution moniker only applies to waves is added in the introduction. Indeed, it's an important point that these traditional coupled "high-resolution" models are forced with atmospheric models with resolutions that are too low to assess tropical cyclone track and core features.

34. There is no explanation about what Cd model used in this study. Although the authors show wind stress values in Figure 10, these values are strongly affected by Cd models or formula, and each wave models may use different Cd model. Need explanation.

    a. Thank you. The text is updated to reflect that the coefficients of surface drag in the "high-resolution" models are from the Charnock formulation (line 103).

35. It is not clear that the meaning of "higher-than-average tropical cyclone activity". If it means that annual occurrence is higher than usual, I comment that it is not affect extreme wave height because the authors use only annual maxima.

    a. This is based on multi-decadal assessment. See the histogram below, for example, for the Mid-Atlantic region. This figure was not originally included in the manuscript for brevity, and can be included if considered important.

[Figure]

*Figure 5: number of tropical cyclone events at the Mid-Atlantic location, 1924 - 2020*

36. The meaning of sentence "in fact..." is not clear. Are extreme distributions in Figure 11a and 12a based on "original high-resolution data set" or "post-processed highresolution data set"? If the result mentioned in the sentence is not shown in graph, it is better to add word "not shown in graph" in the text.
    a. Thank you. This sentence refers to the difference between the solid blue lines and the red dashed lines in revised Figure 5; this reference has been added to the text.

37. It is suggested to draw annual maxima use for fitting of each extreme distribution to understand reasonability, trend etc. of each distribution.
    a. These plots have been added to the revised manuscript in Figure 4.

38. "ERA5-boundary conditions" -> "ERA5 wave boundary conditions" is suggested.
    a. Thank you, this has been updated.

39. It is understood that these four are main conclusions of this study, however, 1) it is questionable that how "200km or less" in bullet 1 is quantified (e.g. cyclone has radius more then 200km exist.)? There is no detail discussion or deep insight about this. 2) bullet 2-4 are already explained in IEC 61400-1 Annex J.  Although the standard is about only for wind, not mentioned for wave, the authors should at least explain and refer in somewhere in this manuscript, then have to explain the differences or originality of this paper.
    a. The 200km threshold was selected as an engineering rule of thumb based on the sizes of hurricanes in the mid-Atlantic and north Atlantic sites. It is certainly true that there are larger tropical storms than this. However, as no distinction is currently made in standards/industrial practices between inside/outside of storm fetch, a value is proposed as a first step.
    b. Local wave features are due to a combination of wind forcing and additional factors, therefore this investigation is considered separate from and in addition to the implications of IEC 61400-1 Annex J. Annex J of IEC 61400-1 covers Monte-

Carlo simulations (synthetic hurricane modeling), and indeed it does identify the separated analysis of storm types. This has not been followed in a number of publications—i.e., use of datasets such as ERA5 and CFSR for determining extremes. However, this is not specified for the wave and ocean environment, and a key motivation of this paper is: given that wave growth is driven by a number of factors, does that matter? We suggest in this paper that it does, and more work is required to quantify which features are important for future ocean modeling and extreme events, be it directly hindcasted or by coupling to synthetic wind fields. Many publications to date use reanalysis datasets that are time-limited to 30-40 years (i.e., CFSR-, HYCOM-derived) and this caveat has implications for use of these datasets for extremal analysis. The authors have not seen mention of this time duration as insufficient for hurricanes but sufficient for winter storms in the standard referenced. As obvious as it may seem, there do not appear to be any standards/requirements that ocean extremes be quantified by the same wind fields as those used to determine wind extremes. This paper is an attempt to highlight some of these gaps in current offshore wind metocean methods. Indeed, there are plenty of public opinions that the current "high-resolution" model approach remains the state of the art for offshore wind design on the US east coast, including when these results were presented at the NAWEA conference.

Finally, the paper lays out the metocean models and analysis process in use in the offshore wind industry today for wider scrutiny, discussion, and replication. Normally, this is not presented in the public domain.

40. What value are used for threshold "u"? Need explanation.
   a. The values of threshold for each analysis are added to legend in revised Figure A2, Appendix A.

[Figure]

[Figure]

(a) Return values at the North Atlantic site from the high-resolution model ("Combined") and two post-processed subsets of the high-resolution model based on storm type.

(b) Return values at the Mid-Atlantic Bight site from the high-resolution model ("Combined") and two post-processed subsets of the high-resolution model based on storm type.

**Figure A2.** Return values by Peaks-Over-Threshold for the "high-resolution" return estimates with a Generalized Pareto distribution and selected thresholds, $u$.

41. Unit for RMSE is needed. Also, definition of NRMSE is needed.
   a. RMSE unit of m is added. NRMSE is changed to Scatter Index, SI. The figures have been replaced with tabulated values from multiple buoys.

42. Because the authors decided to use annual maxima for fitting of extreme value distribution, the validation should be made for annual maximum wave heights. Otherwise, readers don't understand the reasonability of all results and conclusions.

    a. Thank you. Please see the K-S test results in the reply to Question 2, and the distribution fit in reply to Question 1 for goodness of fit assessments of the annual maxima data from each model. Nearby observations to the NA site (NDBC 44097) cover 10 years of data, which is not considered a long enough period to conduct extreme value analysis. However, quantile-quantile plots of annual maxima values between observations and model data, at both sites (note the 40-50km distance between), are presented below for the high-resolution model datasets. Note that due to the differences in dataset duration, some model annual maxima events did not occur during the measurement period.